

# Comparison of artificial neural networks and reservoir models for simulating karst spring discharge on five test sites in the Alpine and Mediterranean regions

Guillaume Cinkus[1], Andreas Wunsch[2], Naomi Mazzilli[3], Tanja Liesch[2], Zhao Chen[4], Nataša Ravbar[5], Joanna Doummar[6], Jaime Fernández-Ortega[7], Juan Antonio Barberá[7], Bartolomé Andreo[7], Nico Goldscheider[2] and Hervé Jourde[1]

[1]HydroSciences Montpellier (HSM), Univ. Montpellier, CNRS, IRD, 34090 Montpellier, France
[2]Karlsruhe Institute of Technology (KIT), Institute of Applied Geosciences, Kaiserstr. 12, 76131 Karlsruhe, Germany
[3]UMR 1114 EMMAH (AU-INRAE), Université d'Avignon, 84000 Avignon, France
[4]Institute of Groundwater Management, Technical University of Dresden, 01062 Dresden, Germany
[5]ZRC SAZU, Karst Research Institute, Titov trg 2, 6230 Postojna, Slovenia
[6]Department of Geology, American University of Beirut, PO Box 11 - 0236/26, Beirut, Lebanon
[7]Department of Geology and Centre of Hydrogeology, University of Málaga (CEHIUMA), 29071 Málaga, Spain

*Correspondence to:* Guillaume Cinkus (guillaume.cinkus@umontpellier.fr)

**Abstract.** Hydrological models are widely used to characterise, understand and manage hydrosystems. Data-driven models are of particular interest in karst environments given the complexity and heterogeneity of these systems. There is a multitude of data-driven modelling approaches, which can make it difficult for a manager or researcher to choose. We therefore conducted a comparison of two data-driven modelling approaches: artificial neural networks (ANN) and reservoir models. We investigate five karst systems in the Mediterranean and Alpine regions with different characteristics in terms of climatic conditions, hydrogeological properties and data availability. We compare the results of ANN and reservoir modelling approaches using several performance criteria over different hydrological periods. The results show that both ANN and reservoir models can accurately simulate karst spring discharge, but also that they have different advantages and drawbacks: (i) ANN models are very flexible regarding the format and amount of input data, (ii) reservoir models can provide good results even with short calibration periods, and (iii) ANN models seem robust for reproducing high-flow conditions while reservoir models are superior for reproducing low-flow conditions. However, both modelling approaches struggle to reproduce extreme events (droughts, floods), which is a known problem in hydrological modelling. For research purposes, ANN models have shown to be useful to identify recharge areas and delineate catchment, based on insights into the input data. Reservoir models are adapted to understand the hydrological functioning of a system, by studying model structure and parameters.



## 1    Introduction

Karst systems are complex and heterogeneous media. High contrasts in porosity and permeability induce a high variability in infiltration and internal flow processes (Bakalowicz, 2005; Ford and Williams, 2007) which can be difficult to assess.

Considering the increasing demand for water and that around 9 % of the world's population (up to 90 % in some parts of the Mediterranean area) depends on karst water resources for drinking water supply (Stevanović, 2019), the characterisation of karst systems functioning and water availability become a major challenge for water resource management. Among the numerous methods to study karst systems (Goldscheider, 2015), hydrological models are useful to characterise karst functioning, and specially to predict the impact of climate and land use changes (Hartmann et al., 2014). Hydrological

models can be grouped into data-driven and distributed approaches (Kovács and Sauter, 2007). While distributed models divide a karst system into a two- or three-dimensional grid, for which each cell is assigned appropriate hydraulic parameters and system states, data-driven models are based on the mathematical analysis of input data (e.g. precipitation, temperature) for simulating spring discharge time series. They include (i) "black-box" models such as neural networks-based approaches, which use no a priori information about the functioning of a system; and (ii) reservoir models, which are based on a

conceptual representation of a karst system – a succession of one or several reservoirs using simplified physical transfer functions.

The choice of a modelling approach depends mainly on the objective of the study, but also on the current knowledge of the system and the available data. For karst systems, the available data are often scarce and poorly reflect the heterogeneity of the meteorological and karst processes. Distributed models require a lot of data for defining physical parameters and thus can

be tough to use in a scarce data context. On the other hand, data-driven models permit studying complex and heterogeneous karst systems without requiring extensive meteorological and system-related data. Both "black-box" and reservoirs models are therefore relevant for operational and research applications. Artificial neural networks (ANN) have been successfully used to simulate karst spring discharge (Kurtulus and Razack, 2007; Hu et al., 2008; Meng et al., 2015; Wunsch et al., 2022), predict and forecast water flood/inrush (Wu et al., 2008; Kong A Siou et al., 2011) and manage the exploitation of karst

aquifers (Yin et al., 2011; Kong A Siou et al., 2015). Reservoir models also have been successfully used to simulate karst spring discharge (Fleury et al., 2007; Dubois et al., 2020), manage the exploitation of karst aquifers (Fleury et al., 2009; Zhou et al., 2021), as well as characterise specific functioning in karst systems (Tritz et al., 2011; Perrin et al., 2003; Bittner et al., 2020; Jukić and Denić-Jukić, 2009). Although several authors compared the performance of different ANN models (Cheng et al., 2020; Kurtulus and Razack, 2010; Kovačević et al., 2018) and studied structure and parameters' equifinality in

reservoir models (Hartmann et al., 2012; Gondwe et al., 2011; Mazzilli et al., 2012; Makropoulos et al., 2008), only few studies have been conducted on the comparison of both approaches (Jeannin et al., 2021; Kong A Siou et al., 2014; Sezen et al., 2019). Kong A Siou et al. (2014) observed that ANN models are more effective for accounting for the nonlinearity of karst systems during extreme events (dry and flood periods), while reservoir models were better for representing the hydrological functioning of the system during intermediate water periods. Sezen et al. (2019) observed that ANN models



were better for simulating low-flow periods and reservoir models for simulating spring discharges on predominantly non-karst catchments. Jeannin et al. (2021) emphasised the great potential of ANN models but highlighted two main limitations: (i) they require long time series to accurately learn the functioning of a karst system, and (ii) usually no information about specific functioning of a system can be deduced from the results.

The performance of ANN and reservoir models can therefore be influenced by the characteristics of the catchment, as well as
the format and length of the input data. The aim of the present study is to help researchers and stakeholders to choose between ANN and reservoir modelling approaches for simulating karst spring discharge, depending on their purpose and the available data. This research provides the first extensive comparison of ANN and reservoir models in karst hydrology by investigating results on five study sites with different context and input data. We use ANN as they have proven to be fast and reliable for modelling hydrological time series (Wunsch et al., 2021; Van et al., 2020; Jeannin et al., 2021). Reservoir
modelling is carried out using the KarstMod platform, as it provides a powerful modular interface for varying the structure, parameters and transfer functions of the conceptual model (Mazzilli et al., 2019). This research seeks to address the following research questions:

- What are the advantages and drawbacks of ANN and reservoir models in karst hydrogeology?
- To which extent can ANN and reservoir models be used to get a better understanding of system functioning?
- What are the implications from a stakeholder's perspective?
- Is one approach better suited for climate change predictions?

## 2   Data and study sites

We compare ANN and reservoir modelling approaches using data from five different well-studied karst systems (Table 1, Fig. A1). All systems have different characteristics in terms of hydrogeological properties (e.g. catchment area,
karstification), data availability (e.g. length of the time series, number of meteorological stations, time step), and environmental conditions (e.g. climate, anthropogenic influence). Each study site is detailed in the following subsections and further details about the meteorological data can be found in Table B1.



**Table 1: Summary of studied springs and areas. $Q_{mean}$ corresponds to the mean observed discharge and $P_{an}$ to the annual mean precipitation over the considered period.**

| Spring | Country | Climate | Catchment area [km$^2$] | $Q_{mean}$ [m$^3$.s$^{-1}$] | $P_{an}$ [mm] | Period |
|---|---|---|---|---|---|---|
| Aubach | Austria | Cool temperate and humid | 9 | 0.91 | 2113 | 2012-11-20 – 2020-10-31 |
| Gato Cave | Spain | Mediterranean | 69-79 | 1.50 | 1872 | 1963-10-02 – 2015-04-29 |
| Lez | France | Mediterranean | 130 | 0.84 | 904 | 2008-10-21 – 2020-12-03 |
| Qachqouch | Lebanon | Mediterranean | 56 | 2.01 | 1293 | 2015-09-06 – 2020-02-05 |
| Unica | Slovenia | Moderate continental | 820 | 21.97 | 1605 | 1961-01-02 – 2018-12-31 |

## 2.1 Aubach spring, Austria

Aubach spring (1080 m asl) is a large non-permanent spring located in the Hochifen-Gottesacker area, on the border between Germany and Austria (Northern Alps). The Hochifen-Gottesacker system covers an area of about 35 km$^2$ and its altitude varies between 1000 and 2230 m asl (Chen et al., 2018). The area is under a cool temperate, humid climate and is strongly affected by snow accumulation and melting, which typically occur between November and May (Chen et al., 2018). The spring is located in the Schwarzwasser valley, which follows the geological contact between highly karstified limestone (Schrattenkalk formation) in the northern and western part and impermeable sedimentary rocks of the Flysch zone in the southern part (Goldscheider, 2005). The main catchment of Aubach spring is estimated to be approximately 9 km$^2$ (Goldscheider, 2005; Chen and Goldscheider, 2014). The spring also receives inflow from several upstream karst catchments and the Flysch zone, where surface runoff can sink into an estavelle and pass through an underground karst conduit during low-flow periods, as demonstrated by multiple tracer tests (Goldscheider, 2005).

Precipitation and temperature data were obtained from three meteorological stations located outside the catchment. The potential evapotranspiration is calculated using data from one station with the modified Turk-Ivanov approach after Wendling and Müller (1984), described in Conradt et al. (2013).

## 2.2 Gato Cave spring, Spain

Gato cave spring (462 m asl) is one of the main outlets of the karst system of Sierra de Lìbar. It is located in the north-western part of the province of Málaga, within the boundaries of the Grazalema Natural Park, about 75 km west of Málaga. The altitude of the Sierra de Lìbar varies between 400 and 1400 m asl according to the main north-east/south-west mountain alignments. The area is under a Mediterranean climate, with an average annual precipitation of about 1500 mm and defined by a strong seasonal pattern (Andreo et al., 2006). The site is located within the External Zone of the Betic Cordillera and presents mainly Jurassic limestones and dolomites, Cretaceous-Paleogene marly-limestones and Tertiary clays and sandstones (Flysch) that cover the whole Mesozoic rock sequence. The Jurassic rocks outcrop as anticlinal cores, while the



synclines and tectonic grabens are composed of Cretaceous rocks (Martín-Algarra, 1987). The Hundidero-Gato system constitutes a binary karst system where a wide range of well-developed karst landforms are found, such as karrenfields,

swallow holes and caves. These features strongly condition recharge, which is primarily produced in two ways: (i) autochthonous, by direct infiltration of rainfall through carbonate outcrops (20-40 km$^2$) as well as rainwater that infiltrates through swallow holes in poljes; and (ii) allochthonous, as a contribution from runoff produced in the Gaduares River basins (43.5 km$^2$). This runoff is stored in the Montejaque dam, which was built on karstified limestone, resulting in water losses in the reservoir and, consequently, the artificial recharge of the aquifer through the Hundidero cave (Andreo et al., 2004).

Precipitation and temperature data are from the meteorological station of Grazalema, which is the closest to the catchment, and therefore the most representative. Potential evapotranspiration is calculated with the Hargreaves-Samani approach (Hargreaves and Samani, 1985).

### 2.3 Lez spring, France

The Lez spring (64 m asl) is located 15 km north of Montpellier, and the altitude of its catchment varies between 64 and 655

m asl. The Lez catchment is exposed to a Mediterranean climate, which is characterised by hot, dry summers, mild winters and wet autumns. As a large part of the hydrogeological basin is relatively impermeable due to the presence of marl and marly-limestone formation, the effective recharge area of the Lez spring covers about 130 km$^2$ (Fleury et al., 2009) and corresponds to Jurassic limestone outcrops. Localized infiltration occurs through fractures and sinkholes along the basin and through the major geologic fault of Corconne-Les Matelles. The Lez aquifer is subject to anthropic pressure (i.e. exploitation

for water supply) with pumping directly into the karstic conduit. The discharge is measured at the spring pool and is regularly zero during low water periods, when the pumping rate exceeds the natural discharge of the spring.

Precipitation data are from four meteorological stations. Three are located in the catchment and one is located about 5 km west of the catchment. Potential evapotranspiration is calculated with the Oudin approach (Oudin et al., 2005). Temperature data are from the Prades-le-Lez meteorological station.

### 135 2.4 Qachqouch spring, Lebanon

Qachqouch spring (64 m asl) is located in the Nahr el-Kalb catchment and originates from a Jurassic karst aquifer. The recharge area is estimated to be about 56 km$^2$ with altitudes ranging from 60 to over 1500 m asl (Doummar and Aoun, 2018; Dubois et al., 2020). The catchment is primarily exposed to a Mediterranean climate, with snow influence at higher altitudes (Dubois et al., 2020). The lithology mainly consists in Jurassic karstified limestone and dolomitic limestone (on the higher

plateaus) changing to more massive micritic limestone in the lower part of the catchment. The Qachqouch system is characterised by a duality of flow in a low permeability matrix and a high permeability conduit system (Dubois, 2017). Potential runoff inflows from higher altitudes and infiltrates downstream into the Jurassic karst aquifer.

Precipitation and temperature data are from two meteorological stations. One is located in the catchment at 950 m asl. The other, with a heated rain gauge, is located 22 km north-east of the catchment at 1700 m asl (Doummar et al., 2018). Potential



evapotranspiration is calculated using data from the *950 m* station with the modified Penman-Monteith approach (Allen et al., 1998).

## 2.5 Unica springs, Slovenia

Unica springs (450 m asl) are the outlets of a complex karst system with an estimated recharge area of about 820 km$^2$. The area is under a moderate continental climate and is strongly influenced by snow accumulation and melting. It is subdivided

into three subcatchments, with a predominance of (i) allogenic infiltration from two subcatchments drained by sinking rivers flowing through a chain of karst poljes and a river valley, and (ii) autogenous infiltration through a karst plateau with highly karstified limestone (Gabrovšek et al., 2010; Kovačič, 2010; Petric, 2010). The poljes follow each other in a descending series at altitudes between 450 and 750 m asl and are connected in a common hydrological system. Characterised by a network of surface rivers and frequent flooding, this induces a very particular response at the Unica springs with very high

hydrological variability (by several orders of magnitude), as well as delayed and prolonged high-flow values (Mayaud et al., 2019). Low-flow periods are sustained by flows from the karstified limestone aquifer, which reaches heights up to 1800 m asl and has significant groundwater storage (Ravbar et al., 2012). Part of the discharge is lost due to an underground bifurcation (Kogovšek et al., 1999). When the discharge exceeds about 60 m$^3$ s$^{-1}$ and remains high for a few days, a polje downstream of the springs becomes flooded. When the discharge reaches about 80 m$^3$ s$^{-1}$, the flooding reaches the

monitoring station, influencing the measurement. The water from the lake is drained by several ponors downstream of the monitoring station, but their absorption capacity is much lower than the discharges of the springs.

Precipitation, snow cover height, and height of new snow data were obtained from two meteorological stations located on the catchment. Temperature and relative humidity data are from Postojna meteorological station only. Potential evapotranspiration is calculated using data from the Postojna station with the modified Penman-Monteith approach (Allen et

al., 1998).

## 3 Methodology

### 3.1 Artificial neural networks

ANN are a branch of Machine Learning, i.e. a technique to learn complex relations from existing data. They imitate the basic functioning of biological nervous systems and similarly consist of mathematical representations of neurons structured and

interconnected in layers. Given sufficient data from which to learn, ANN can establish complex input-output relations with only limited domain knowledge.

In this study, Convolutional Neural Networks (CNN) (LeCun et al., 2015) – a specific model type from the ANN-subfield of Deep Learning (DL) – is used. CNN are predominantly successful in processing image-alike data, but are also useful in signal processing for sequential data. They usually consist of sequences or blocks of convolutional layers for feature

recognition and pooling layers for information consolidation. In the former, filters of a specific size (defining their receptive





field) are used to produce feature maps. These feature maps are subsequently down-sampled (often by maximum selection) into pooling layers to consolidate the contained information. Several of these blocks with varying properties can be stacked on top of each other, also in combination with other layer types such as batch normalization layers (Ioffe and Szegedy, 2015) to prevent exploding gradients or dropout layers (Srivastava et al., 2014). Lastly, one (or multiple) fully connected dense

layers follow to produce the model output. For the models in this study, we used a single 1D-Convolutional layer with a fixed kernel size (three) and an optimised number of filters. This layer was succeeded by (i) a Max-Pooling layer, (ii) a Monte-Carlo dropout layer (10 % dropout rate) and (iii) two dense layers: the first with an optimised number of neurons and the second with a single output neuron. We programmed our models in Python 3.8 (van Rossum, 1995), using the following frameworks and libraries: Numpy (van der Walt et al., 2011), Pandas (Reback et al., 2021; McKinney, 2010), Scikit-Learn

(Pedregosa et al., 2018), Matplotlib (Hunter, 2007), BayesOpt (Nogueira, 2014), TensorFlow 2.7 (Abadi et al., 2016) and its Keras API (Chollet et al., 2015).

## 3.2 Reservoir models

Reservoir models are a conceptual representation of a hydrosystem, which involves the association of several reservoirs that are thought to be representative of the main processes at stake. Each reservoir is characterised by its water height and a flow

equation that translates the variations of water height into discharges. The flow equation is function of a specific discharge coefficient and a positive exponent (different from 1 for non-linear flows), which are defined by calibration against observed data.

Many reservoir models have been developed to study the relation between precipitation and discharge in karst systems (Hartmann et al., 2014). They all differ in complexity with respect to the number of reservoirs and parameters, which need to

be well thought out in order to preserve physical realism and limit equifinality on model parameters. Careful sensitivity analyses and uncertainty assessment should be considered along with model results to avoid over-interpretation (Refsgaard et al., 2007). Reservoir models can be seen as a compromise between simulation performance and insight into the functioning of a system. This approach is well suited to karst systems due to the high heterogeneity and low level of knowledge of their structure (Hartmann et al., 2012; Fleury et al., 2009).

We used the adjustable modelling platform KarstMod for performing reservoir modelling. KarstMod provides a modular, user-friendly interface for simulating spring discharge at karst outlets. Structure of models built using KarstMod platform is based on the conceptual model of a karst aquifer with infiltration and saturated zones (Mazzilli et al., 2019). The infiltration zone (soil and epikarst) drains water from the surface through a vertical network of fissures and conduits. Water storage can occur in the unsaturated zone, as well as local saturation. The saturated zone comprises a dual porosity functioning, with a

network of high-permeability fractures and conduits, and a low-permeability matrix with a high storage capacity.

In KarstMod, the model structure can include up to four reservoirs. One at the upper level reflects the processes (infiltration, storage and drainage) occurring in the soil and epikarst zone. Three at the lower level can be connected with the first one and correspond to the infiltration and/or saturated zones. The discharge can be simulated with (i) several linear and non-linear





water level-discharge laws, (ii) a hysteretic water level-discharge function to reproduce the hysteretic functioning observed

on the wet-dry cycles in the unsaturated zone (Lehmann et al., 1998; Tritz et al., 2011), and (iii) an exchange function that aims to reproduce the interactions between matrix and conduits. More details on the balance equations, the parameters involved and the KarstMod platform in general can be found in Mazzilli et al. (2019) or in the KarstMod User Guide (Mazzilli and Bertin, 2019).

In this study, we first addressed the structure of the models taking into account our expert knowledge and previous studies.

For each site, we examined the major characteristics that determine the functioning of the system and associated the corresponding conceptual modelling. We then modified this base structure according to the performance of the model while trying to maintain physical realism. The most efficient model structures that we obtained after performing the modelling are shown in Fig. 1.

Aubach spring selected model (Fig. 1a) is close to the conceptual model with a very reactive transfer function ($Q_{ES}$),

corresponding to the well-developed conduit network, and a matrix reservoir (M), which in this case mostly reflects the storage properties in the unsaturated limestone. We tested different configurations (lost discharge from upper level reservoir and/or pumping in lower reservoirs) to simulate the lost discharges through overflow springs and underground flows, but there were no significant increases in model performance. Gato Cave spring selected model (Fig. 1b) is different from the conceptual model as the platform could not account for the allochthonous recharge on the catchment. The model structure

includes a soil available water capacity ($E_{min}$), matrix and conduits compartments (M and C), as well as matrix-conduits exchanges ($Q_{MC}$), which may translate the processes occurring through the dam. Lez spring selected model (Fig. 1c) is accurate with the conceptual model and includes an overflow transfer function ($Q_{loss}$), matrix and conduits compartments (M and C), matrix-conduits exchanges ($Q_{MC}$), and pumping into the main conduit ($Q_{pump}$). We considered a low soil available water capacity ($E_{min}$) as it greatly increased the performance of the model. Qachqouch spring selected model (Fig. 1d) is

consistent with previous conceptual models that considered many different response times. The model structure features a very reactive transfer function ($Q_{ESO}$), matrix and conduits compartments (M and C), matrix-conduits exchanges ($Q_{MC}$) as well as a soil available water capacity. The multiple different transfer functions help to reproduce the reactive and dampened responses of the Qachqouch karst aquifer. Unica springs selected model (Fig. 1e) is significantly simpler than the conceptual model, which includes polje flooding, allochthonous recharge, overflow springs and matrix-conduits exchanges. We only

retained a very simple structure as it was the best trade-off between physical realism and model performance. The very reactive transfer function $Q_{ESO}$ allows reproducing fast flows through conduits, while the matrix reservoir (M) likely translates processes occurring in the matrix and surface flooding.





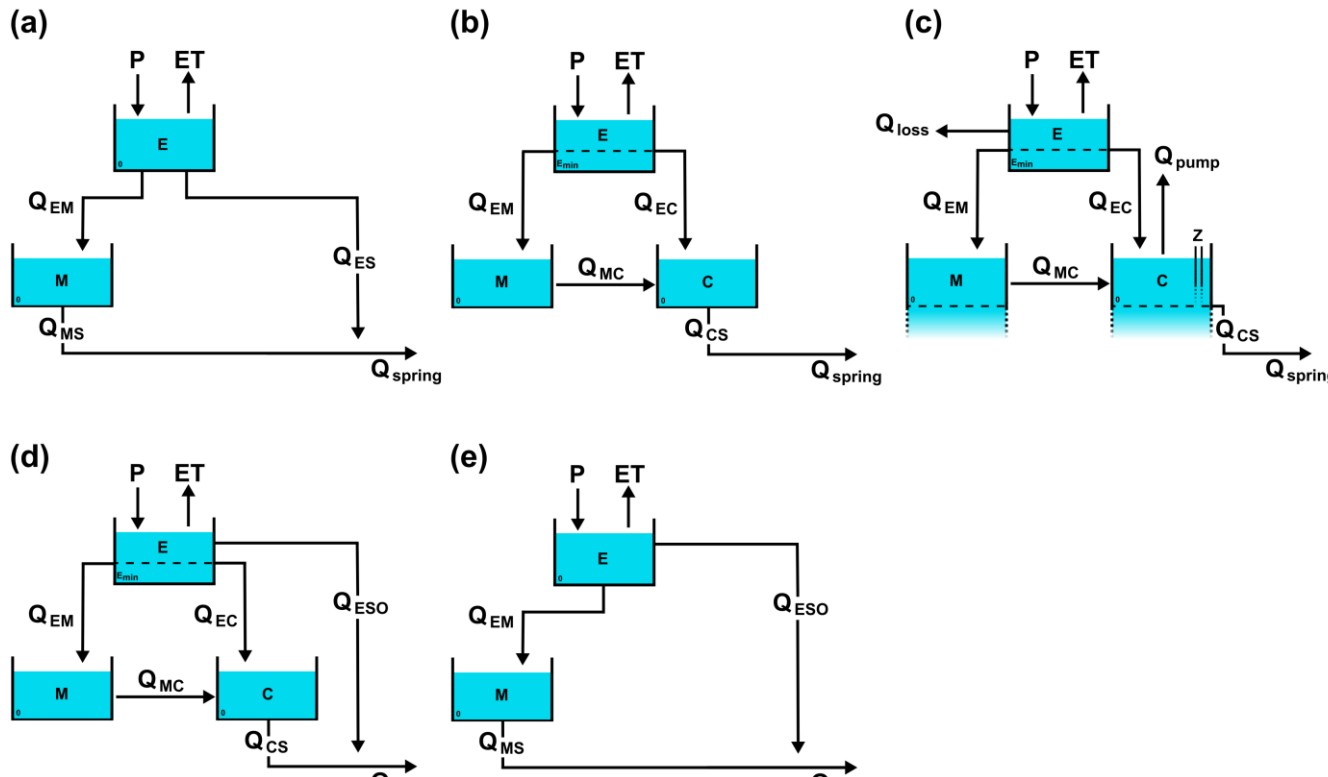

**Figure 1: Selected model structures for (a) Aubach, (b) Gato Cave, (c) Lez, (d) Qachqouch and (e) Unica springs. Flux names**
**correspond to the terminology of the KarstMod platform (Mazzilli and Bertin, 2019).**

### 3.3    Input data

Input data are the time series that are used for simulating karst spring discharge. They can be derived from either a direct observation (e.g. observed discharge, temperature, sinking stream discharge or pumping) or a calculation from raw input data (e.g. potential evapotranspiration derived from temperature). Nature of input data usually differs between ANN and reservoir
modelling approaches, as ANN models tends to make good use of direct observations, whereas reservoir models often requires to preprocess the raw input data. We decided to work with raw input data to ensure equitable performance between ANN and reservoir models. The raw input data was either used directly or preprocessed, depending on the modelling approach.

The data used for each modelling approach and site is summarised in Table 2. Observed discharge time series were used
directly (without further preprocessing) in ANN and reservoir models. In the case of the Lez spring, the models were simultaneously calibrated on the spring discharge (Q) as well as on the water level in the aquifer (Z). Furthermore, the pumped discharge time series in reservoir C ($Q_{pump}$, Fig. 1c) was used as an input. Precipitation time series were used differently as there are often several meteorological station per study site. For ANN models, precipitation time series were used as raw input $P_{raw}$, except for Lez spring where the individual raw precipitation data had too many missing values so we



used the same input as the reservoir model ($P_{in}$). In the case of Aubach, Qachqouch and Unica, $P_{raw}$ includes all the precipitation time series from the different meteorological stations (Table B1). For reservoir models, the precipitation time series were either (i) used directly if there was no snow dynamics on the catchment and only one meteorological station was available (Gato Cave), (ii) preprocessed with Thiessen's polygons interpolation (Appendix C) if there were several meteorological stations (Lez), (iii) preprocessed with a snow routine (Appendix D) to simulate snow accumulation and melting over the catchment (Aubach) if snow dynamics could not be neglected, or (iv) preprocessed with both Thiessen's polygons interpolation and snow routine (Qachqouch, Unica). For reservoir models, evapotranspiration processes were considered using time series of potential evapotranspiration. For ANN models, we used temperature time series instead of evapotranspiration because calculating potential evapotranspiration is generally not necessarily beforehand. Additionally, we used a sinusoidal temperature signal time series ($T_{sin}$, derived from the observed temperature) to better account for seasonality in Aubach, Lez and Unica ANN models.

**Table 2: Summary of input data. (i) $P_{raw}$, (ii) $P_{in}$ and (iii) $P_{sr}$ refer to (i) raw precipitation data, (ii) precipitation data interpolated with Thiessen's polygons method, and (iii) precipitation data redistributed by applying the snow routine. $Q_{obs}$, $Z_{obs}$ and T refer to observed discharge, observed water level and temperature, respectively. ET (Evapotranspiration) refers to either PET (Potential Evapotranspiration) or AET (Actual Evapotranspiration) time series.**

| Spring | Time step | Date Range | Data used | | Maximum gap [days] | | | |
|---|---|---|---|---|---|---|---|---|
| | | | ANN | Reservoir | P | T | Q | ET |
| Aubach | Hourly | 2012–2020 | $Q_{obs}$, $P_{raw}$, T, $T_{sin}$[a] | $Q_{obs}$, $P_{sr}$, PET[c] | 0 | 0 | 0 | 0 |
| Gato Cave | Daily | 1963–2015 | $Q_{obs}$, $P_{raw}$, $T_{max}$, $T_{min}$, $T_{med}$ | $Q_{obs}$, $P_{raw}$, PET | 0 | 0 | 0 | 0 |
| Lez | Daily | 2008–2020 | $Q_{obs}$, $Q_{pump}$, $Z_{obs}$, $P_{in}$, $T_{sin}$ | $Q_{obs}$, $Q_{pump}$, $Z_{obs}$, $P_{in}$, PET | 0 | 2 | 7 | 0 |
| Qachqouch | Daily | 2015–2020 | $Q_{obs}$, $P_{raw}$, $T_{max}$[b] | $Q_{obs}$, $P_{in}$, PET | 0 | 0 | 11 | 0 |
| Unica | Daily | 1961–2018 | $Q_{obs}$, $P_{raw}$, T, $T_{sin}$, NS | $Q_{obs}$, $P_{in-sr}$, PET | 0 | 1 | 0 | 29 |

[a]$P_{raw}$, T and $T_{sin}$ data are from Diedamskopf, Oberstdorf and Walmendinger Horn meteorological stations

[b]$T_{max}$ data are from the 1700 m meteorological station

[c]$P_{sr}$ data are calculated with the data from Diedamskopf station

We handled missing values in the different time series as follows: (i) temperature gaps were linearly interpolated, (ii) precipitation and evapotranspiration gaps were considered to be equal to 0, and (iii) discharge gaps were interpolated with a Lagrange polynomial function. Maximum observed gaps for precipitation, temperature, discharge and evapotranspiration time series are detailed in Table 2. Note that (i) for Lez spring, we observed maximum gaps of 17 and 16 days for pumped discharge and piezometric level, respectively; and (ii) for Unica springs, there are no missing values in the Cerknica new snow height (NS) time series.





### 3.4 Model calibration and simulation

The calibration period is the period used for selecting the parameters that provide the best results according to the performance measure. The validation period is intended to assess the relevance of the parameters over a time interval that is not used for calibration. In the domain of the ANN modelling, the validation is usually denoted as testing period. However, we unify the terminology at this point. The calibration period is again split into three different parts in the case of ANN modelling, (i) to train the model, (ii) to prevent the model from overfitting, and (iii) to optimize its hyperparameters. We defined the same calibration and validation periods for both modelling approaches (Table 3), which ensures fair initial conditions and a meaningful comparison of the results. We have chosen the periods in a way to maximise the length of the calibration periods, which allows for relevant model results (especially in ANN models). In reservoir model, we considered a short warm-up interval at the beginning of the calibration period for the model to adjust and reach an optimal state.

**Table 3: Calibration and validation periods.**

| Spring | Calibration period | Validation period | Objective function | |
|---|---|---|---|---|
| | | | ANN | Reservoir |
| Aubach | 2014-04-18 – 2019-12-31 | 2020-01-01 – 2020-10-31 | MSE(Q) | NSE(Q) |
| Gato Cave | 1963-10-02 – 2011-09-03 | 2011-09-04 – 2015-04-29 | MSE(Q) | NSE(Q) |
| Lez | 2008-10-21 – 2017-12-31 | 2018-01-01 – 2020-12-03 | MSE(Q, Z) | NSE(Q, Z) |
| Qachqouch | 2015-09-06 – 2019-09-30 | 2019-10-01 – 2020-01-22 | MSE(Q) | NSE(Q) |
| Unica | 1961-01-02 – 2016-09-28 | 2016-09-29 – 2018-12-31 | MSE(Q) | NSE(Q) |

We calibrated the models by applying the Mean Squared Error (MSE) and the Nash-Sutcliffe Efficiency (NSE, Nash and Sutcliffe (1970)) criteria – in ANN and reservoir models, respectively – on simulated and observed discharges time series. For Lez spring, we used a composite function of discharge and water level in order to consider both variables in the same modelling process.

A total of 1000 simulations were carried out for each modelling approach at each site. The obtained simulated discharge (or water level) time series corresponds to the mean of the distribution of simulated values at each time step. We also considered the uncertainties in the model prediction by calculating the 90 % confidence interval, whose limits correspond to the 0.05 and 0.95 quantile of the distribution at each time step.

In KarstMod (reservoir models), the simulations correspond to the 1000 best results of a one-hour simulation run. The confidence interval reflects the uncertainty in the parameters used in the model, which are not fixed but defined as a range (e.g. catchment area = 150 to 200 km$^2$). In the case of ANN models, we used a model ensemble of 10 models based on





different random number generator seeds for model initialization. Using the Monte-Carlo dropout layer, for each of the ensemble members a total of 100 simulation results were generated.

### 3.5 Model evaluation

We evaluated the performance of the models using several performance criteria that assess different aspects of the discharge:
modified Kling-Gupta Efficiency (KGE'), KGE' components ($r, \gamma, \beta$) (Kling et al., 2012) and Diagnostic Efficiency (DE) (Schwemmle et al., 2021). These criteria were all applied to the whole discharge regime, but also to sub-regimes of high- and low-flow conditions (with the exception of DE, which already takes sub-regimes into account). For Lez spring, we also applied the KGE' criterion on water level. Model performance is usually evaluated on both calibration and validation periods for reservoir models. However, this approach is not suited to ANN models, for which the calibration period corresponds to
the learning period of the model. Thus, we chose to only evaluate and compare the reservoir and ANN models on their validation periods.

The KGE' is based on the Kling-Gupta Efficiency (Gupta et al., 2009) and aims to ensure that bias and variability are not cross-correlated by using the coefficient of variation ratio ($\gamma$) instead of the standard deviation ratio ($\alpha$):

$$KGE' = 1 - \sqrt{(r-1)^2 + (\gamma-1)^2 + (\beta-1)^2} \qquad (1)$$


With $r$ the Pearson correlation coefficient between the simulated and observed discharge, $\beta$ the ratio between mean simulated and mean observed discharge, and $\gamma$ the ratio between simulated and observed coefficient of variation of discharge. The three components of KGE' help to evaluate different aspects of a model: (i) $r$ is related to shape and timing (Santos et al., 2018), (ii) $\beta$ is used to assess the overall volume of water discharged at the spring (further referred to as
"volume"), and (iii) $\gamma$ gives insight into the flow variability. The KGE' and $r$ criteria can range from -∞ to 1, whereas $\gamma$ and $\beta$ can range from -∞ to ∞. A KGE' score equal to 1 means a perfect match between simulated and observed discharge, while a score lower than -0.41 indicates that the model is worse than using the observed mean as a predictor (Knoben et al., 2019).

The DE criterion is intended to help defining the weaknesses of a model. It is based on constant, dynamic and timing errors.
DE is proposed as a numerical measure (ranging from 0 to ∞, with 0 indicating a perfect model), but can also be visualized on a polar plot that effectively differentiate error contributions. The overall error is calculated with the following equation:

$$DE = \sqrt{\overline{B_{rel}}^2 + |B_{area}|^2 + (r-1)^2} \qquad (2)$$






With $\overline{B_{rel}}$ and $|B_{area}|$ the measures for constant and dynamic errors, respectively. As these measures are based on the flow duration curve, they give an information in terms of exceedance probability. Details of their calculation can be found in Schwemmle et al. (2021).

The performance criteria applied to high- and low-flow conditions are denoted by the lower script indices "L" and "H",

respectively. These criteria allow the performance of the models to be evaluated over different flow regimes (i.e. dry/intermediate, wet). Discharge thresholds were set manually based on our knowledge of the system and a careful assessment of the distribution of discharge values. They are equal to 1, 2, 0.8, 5, and 20 m$^3$ s$^{-1}$ for Aubach, Gato Cave, Lez, Qachqouch and Unica springs, respectively.

## 4    Results and discussion

The obtained models and their confidence intervals for the two approaches and each test site are presented in Fig. 2 for discharge and Fig. 3 for water level (Lez spring). Their performance – assessed with multiple criteria – are shown in Fig. 4 and in Table 4. The DE polar plots for each site are presented in Fig. 5.









**Figure 2: Observed and simulated spring discharge time series with (i) 90 % confidence intervals (CI) on the validation period and**
**(ii) threshold for high and low flows used for the calculation of the performance criteria. (a) Aubach, (b) Gato Cave, (c) Lez, (d) Qachqouch and (e) Unica springs.**

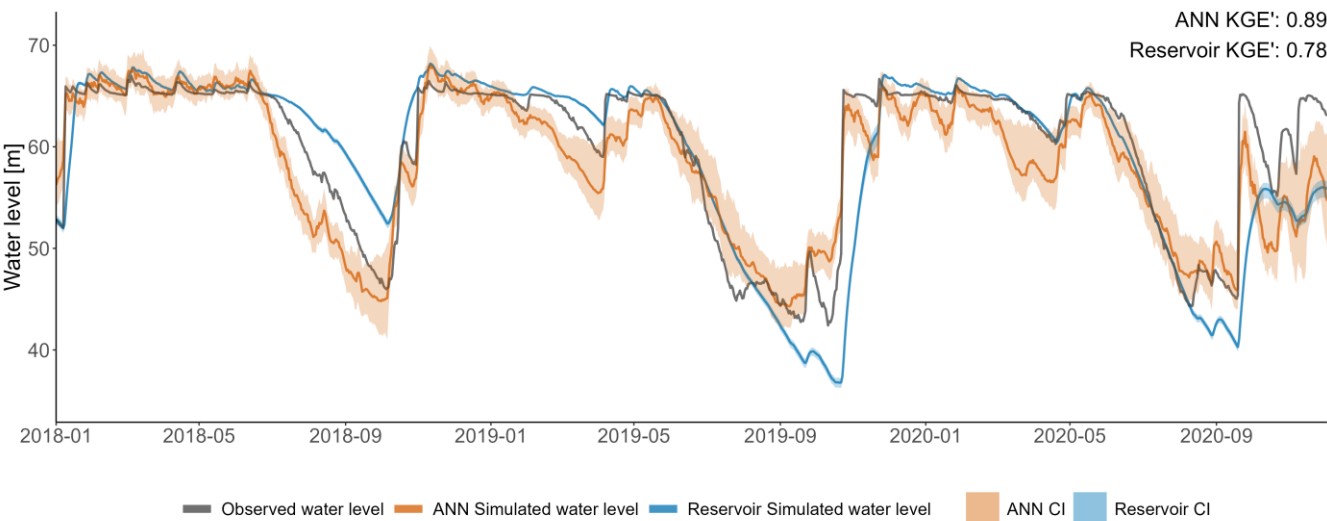

**Figure 3: Observed and simulated spring water level time series with 90 % confidence intervals (CI) on the validation period (Lez spring).**






**Figure 4: Performance of the ANN and reservoir models on the validation period, according to different performance criteria. Exact values can be found in Table 4.**





**Table 4: Details of indicator values for the reservoir and ANN models on the validation period. For each site, the simulations are evaluated with different performance criteria on total, high- and low-flow conditions. Values in bold correspond to the better score between ANN and reservoir models.**

| Spring | Flow regime | KGE' | | β | | r | | γ | |
|---|---|---|---|---|---|---|---|---|---|
| | | ANN | Reservoir | ANN | Reservoir | ANN | Reservoir | ANN | Reservoir |
| Aubach | Total | **0.88** | 0.67 | 0.93 | **0.94** | **0.91** | 0.68 | **1.01** | 1.05 |
| | High flow | **0.80** | 0.47 | **0.91** | 0.85 | **0.84** | 0.54 | **1.07** | 1.22 |
| | Low flow | **0.57** | -0.21 | **0.99** | 1.25 | **0.66** | 0.40 | **1.26** | 2.01 |
| Gato Cave | Total | **0.91** | 0.85 | **0.98** | 0.88 | 0.92 | **0.95** | **0.97** | 0.91 |
| | High flow | 0.77 | **0.79** | **0.92** | 0.82 | 0.82 | **0.90** | 1.11 | **0.98** |
| | Low flow | 0.59 | **0.67** | 1.32 | **1.22** | 0.82 | **0.86** | **1.19** | 1.20 |
| Lez | Total | 0.70 | **0.80** | 0.74 | **0.88** | **0.93** | 0.88 | 1.13 | **1.10** |
| | High flow | **0.52** | 0.49 | 0.75 | **0.87** | **0.84** | 0.67 | 1.38 | **1.36** |
| | Low flow | 0.38 | **0.65** | 0.64 | **0.99** | 0.64 | **0.76** | 1.34 | **1.26** |
| Qachqouch | Total | 0.67 | **0.86** | 0.87 | **1.01** | 0.82 | **0.94** | 0.75 | **0.87** |
| | High flow | 0.22 | **0.57** | 0.71 | **0.96** | 0.46 | **0.89** | 0.51 | **0.59** |
| | Low flow | **0.74** | 0.73 | 1.21 | **1.11** | 0.91 | **0.95** | **1.12** | 1.24 |
| Unica | Total | 0.73 | **0.78** | **1.03** | 0.80 | 0.93 | **0.94** | 0.74 | **0.91** |
| | High flow | **0.73** | 0.69 | **0.87** | 0.75 | 0.79 | **0.82** | 0.89 | **0.99** |
| | Low flow | 0.07 | **0.57** | 1.92 | **1.10** | **0.86** | 0.76 | **0.95** | 1.34 |





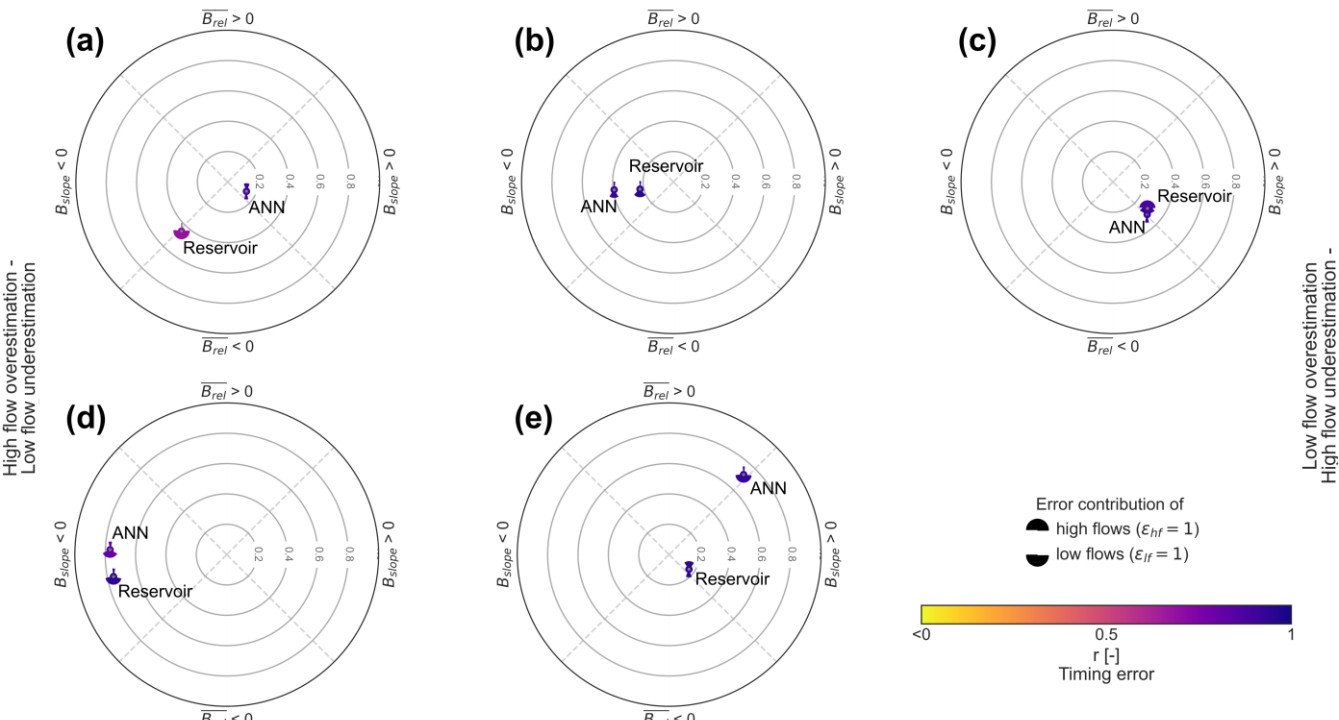

**Figure 5: Diagnostic efficiency polar plots on the validation period. (a) Aubach, (b) Gato Cave, (c) Lez, (d) Qachqouch and (e) Unica springs.**

### 4.1 Modelling results

#### 4.1.1 Aubach spring

ANN model is very good with a KGE' of 0.88 (Table 4). The snow-influenced period from April to mid-June is accurately modelled, as are the peaks in summer and early autumn (Fig. 2a). The highest peaks of the whole time series occurring in February, July and November are only slightly underestimated. The model is balanced and accurate on volume ($\beta$=0.93), variability ($\gamma$=1.01) and shape and timing ($r$=0.91). The model is very good for simulating high flows and is decent on low flows, but could be improved especially on shape and timing ($r_H$=0.84, $r_L$=0.66). The slightly higher value of $\gamma_L$ (1.26) may be related to the tendency of the model to "oscillate" during low/medium flows (e.g. in September, Fig. E1a). This wave-like behaviour may be related to a high sensitivity to precipitation events or to an inappropriate learning from other data. DE is very good (0.14, Fig. 5a). The model shows negative dynamic and constant errors with a higher share of high flows, which points a small underestimation of the occurrence of high flows.

Reservoir model is decent with a KGE' of 0.67 (Table 4), but the model fails to accurately reproduce the discharges in all seasons. There is a deficit in water during winter/early spring and an excess during spring (Fig. 2a). The model is balanced and accurate on volume ($\beta$=0.94) and variability ($\gamma$=1.05), but has middling shape and timing ($r$=0.68). The model is particularly bad for simulating low flows, with high errors on volume ($\beta_L$=1.25), variability ($\gamma_L$=2.01) and shape and timing



($r_L$=0.4). The simulated high flows are decent, although they are also poor on shape and timing ($r_H$=0.54). DE is somewhat insufficient (0.45, Fig. 5a). The model has a positive dynamic error and a negative constant error, with a higher share of low flows, which highlight a substantial underestimation of the occurrence of low flows. These errors are probably due to a miscalibration of the snow routine, retaining too much water as snow in winter and thus releasing too much in warmer periods.

In October, a series of peaks is not well captured by the outputs of both models (Fig. 2a). A plausible explanation is that the inputs do not capture the respective local precipitation events due to the location of the climate stations outside the catchment.

The modelling of discharges from Aubach spring is challenging due to the large elevation differences and the heterogeneity of precipitation on the catchment. This makes it difficult to provide accurate data to the model, especially with regard to snow dynamics. The reservoir model is particularly confronted with these aspects because (i) it can only handle a single precipitation input (from one weather station or interpolated from several stations) and (ii) the snow dynamics must be simulated by a snow module. As these preprocessings are not included in the model calibration and are mostly performed manually, they strongly limit the model performance. Leaving aside the mismatches related to inadequate meteorological inputs, the structure of the reservoir model seems appropriate to simulate the hydrological response of the spring. In contrast, the ANN model is able to consider snow dynamics without any preprocessing, using only the precipitation and temperature time series during calibration. It shows a great versatility with respect to the input data, similar to that of a two-dimensional approach.

### 4.1.2 Gato Cave spring

ANN model is very good with a KGE' of 0.91 (Table 4), but the model struggles to reproduce the discharges during flood events (Fig. 2b). Very high peaks are either overestimated (e.g. May 2012, April 2013, March 2014) or underestimated (e.g. December 2011, November 2012, March 2013). The model is balanced and accurate on volume ($\beta$=0.98), variability ($\gamma$=0.97) and shape and timing ($r$=0.92). The model is good for simulating high flows and is somewhat decent on low flows. It shows a slight lack on shape and timing on both high and low flows ($r_H$=0.82, $r_L$=0.82), and also seems to overestimate low flows ($\beta_L$=1.32). In the same way as Aubach ANN model, the slightly high variability ($\gamma_L$=1.19) may be related to the "oscillations" that can be observed especially on medium and low flows (e.g. January 2012, May 2013, Fig. E1b).

Reservoir model is very good with a KGE' of 0.85 (Table 4), although the model tends to slightly underestimate the discharges during high-flow events ($\beta_H$=0.82; Fig. 2b). This seems to happen when precipitation occur during several days without reaching really high values, which may indicate either (i) some kind of hysteresis functioning with flow occurring after a connection has been made in the system, or (ii) inflows into the system that are not taken into account in the model. The model is balanced and accurate on variability ($\gamma$=0.91) and shape and timing ($r$=0.95), but generally underestimates volume ($\beta$=0.88). The model has good performance on high flows and is decent on low flows. After flood periods, the model seems to simulate a slower draining than observed – higher volume ($\beta_L$=1.22) and variability ($\gamma_L$=1.2) of low/medium flows



– resulting in inaccurate recession periods for which the discharge is overestimated (e.g. November 2011, January 2013,
April 2014).

Some periods like November 2012 or February 2015 are not simulated very well by both models (Fig. 2b), which may be
related to uncertainties in the meteorological data input. DE is decent (0.39) for ANN model and good (0.23) for reservoir
model (Fig. 5b). Both models have a positive dynamic error with a higher share of low flows, which highlight a small
underestimation of the occurrence of low flows.

The modelling of discharges from Gato Cave spring shows that both approaches can have great performance given few
modelling constraints. Raw precipitation input was used in both models, therefore avoiding additional uncertainties from
interpolation or snow preprocessings.

### 4.1.3    Lez spring

ANN model is good with a KGE' of 0.7 for discharge (Table 4) and 0.89 for water level (Fig. 3). The high piezometric levels
(above 55 m asl) seems a bit too sensitive to precipitation events, especially at the end of 2019 (Fig. 2c). On discharges, the
model is accurate on variability ($\gamma$=1.13) and shape and timing ($r$=0.93), but underestimates volume ($\beta$=0.74). The overall
underestimation of volume mainly comes from high flows ($\beta_H$=0.75) as they are the most represented on the time series. The
model is decent on high flows, although having too much variability ($\gamma_H$=1.38). On low flows, the model performs poorly
mainly due to high underestimation of volume ($\beta_L$=0.64) and insufficient shape and timing ($r_L$=0.64).

Reservoir model is very good with a KGE' of 0.8 for discharge (Table 4) and 0.78 for water level (Fig. 3). However, the
model fails to reproduce the observed discharge for several months for the period between September 2017 and March 2018
(Fig. 2c). During dry periods, there is a too high deficit in the lower reservoirs, leading to a strong delay in the spring
response when fresh precipitation occur – the C reservoir having to be replenished beforehand. The model is balanced and
accurate on variability ($\gamma$=1.1) and shape and timing ($r$=0.88), but underestimates volume ($\beta$=0.88). The model is decent on
high flows, but has poor variability ($\gamma_H$=1.36) and shape and timing ($r_H$=0.67), and also slightly underestimated volume
($\beta_H$=0.87). On low flows, the model has too much variability ($\gamma_L$=1.26) and middling shape and timing ($r_L$=0.76). The
piezometric levels are satisfactory when the spring is flowing (greater than 65 m asl), but lose accuracy during dry periods.
The model could not reproduce the changes in flow dynamics at 46 m asl (August 2019, August 2020, Fig. 3). Also, the
draining of the aquifer seems to be simulated slower than observed (July 2018, July 2019), which can be a result of the
model trying to fit the aforementioned periods during calibration.

On both models, the poor overall KGE' value on low/medium flows is probably due to the small occurrences of low
discharges (except 0), thus inducing a high error on volume. DE is good for both ANN (0.31) and reservoir models (0.29)
(Fig. 5c). Both models have negative dynamic and constant errors with a higher share of high flows, which highlight an
underestimation of the occurrence of high flows.

The time series is generally characterised by distinct dry periods without any recharge due to the anthropogenic pumping of
water into the saturated zone of the aquifer. These periods are simulated fairly accurately by both models but ANN model is





better at simulating first floods after or during dry periods. Several boreholes at the north of the spring showed flow-bearing structures at 50 m asl (Dausse et al., 2019). These fast water transfer could explain the rapid increases in observed piezometric level and the reactive spring responses. We also suspect an evolution of the carbonate's facies with depth, which

could affect the effective porosity of the medium and induce different flow dynamics. These aspects are not considered in the reservoir model which results in poor simulations when the water level is below 60 m asl. However, this failure provides information on the structure of the aquifer, which is valuable for research. On the other hand, ANN model was successful in learning these particular behaviours, especially as it only had a medium learning time of about 8 years.

### 4.1.4    Qachqouch spring

ANN model is decent with a KGE' of 0.67 (Table 4), but strongly overestimates low flows at the beginning of December, then underestimates the flood peak at the end of the month (Fig. 2d). The model slightly underestimates volume ($\beta$=0.87), and is lacking in variability ($\gamma$=0.75) and shape and timing ($r$=0.82). The high flows are poorly simulated but the low flows are well fitted, although volume is slightly overestimated ($\beta_L$=1.21). The oscillations of the simulated discharges (Fig. E1c) may appear because the model does not account the time needed for the aquifer to replenish and generate an increase of

discharge at the spring.

Reservoir model is very good with a KGE' of 0.86 (Table 4). At the end of the dry period, the low flows are overestimated and the first flood is underestimated (Fig. 2d). This may be due to heterogeneous precipitation occurring on highly transmissive parts of the catchment. In this case, the soil available water capacity ($E_{min}$) – which is necessary for a good simulation of low-flow periods – may not be representative of the whole catchment, thus inducing a more dampened

response than observed. The model is balanced and accurate on volume ($\beta$=1.01) and shape and timing ($r$=0.94), but slightly lacks in variability ($\gamma$=0.87). The model is decent on high flows but has middling variability ($\gamma_H$=0.59) which can be due to the underestimation of the late December flood peak. The low flows and recession periods are slightly overestimated ($\beta_L$=1.11 and $\gamma_L$=1.24).

DE is bad for ANN and reservoir models (0.77 and 0.76, respectively, Fig. 5d). Both models have a positive dynamic error

with a higher share of low flows, which highlight an underestimation of the occurrence of low flows. Here, the positive dynamic error is influenced by the constant underestimation of the observed discharge during the dry period (October–December 2019), accounting for more than 50 % of the observations.

The very short data length is particularly detrimental to the ANN model as the learning period is only about 3 years. Furthermore, even when data are available, there is a significant amount of time without (relevant) discharge, for which no

input-output relation can be learned. Due to the characteristics of the discharge time series, it can be assumed that a much longer time series of daily values would be needed to successfully simulate the discharges of Qachqouch spring. On the other hand, the reservoir model seems more appropriate to simulate Qachqouch spring discharges even with the limited data available. This highlights the strength of conceptual modelling to take into account recharge processes and reservoir replenishment, even on a short dataset.





### 4.1.5    Unica springs

ANN model is good with a KGE' of 0.73 (Table 4). The model is accurate on volume ($\beta$=1.03) and shape and timing ($r$=0.93), but insufficient on variability ($\gamma$=0.74). The model is good at simulating high flows, although slightly lacking in volume ($\beta_H$=0.87), variability ($\gamma_H$=0.89) and shape and timing ($r_H$=0.79). The model is poor for simulating low flows, which are often significantly overestimated ($\beta_L$=1.92), especially the recession periods which systematically have a slower draining (Fig. 2e). The overestimation of low flows could be the result of the model trying to better fit the high-flow periods during training, which may shift the whole discharge curve slightly towards the upper limits. The model also seems to be too sensitive regarding precipitation events, hence the wave-like behaviour of the simulated time series (Fig. E1d). DE is bad (0.72, Fig. 5e). The model has a negative dynamic error and a positive constant error with a higher share of low flows, which highlights an overestimation of the occurrence of low flows.

Reservoir model is good with a KGE' of 0.78 (Table 4). The model is balanced and accurate on variability ($\gamma$=0.91) and shape and timing ($r$=0.94), but has shortcomings on volume ($\beta$=0.8). In some winter months (December 2017, March 2018), the model has a delayed response of the rising limb (Fig. 2e), which may be due to a slightly wrong parametrisation of the snow routine. The model is good on high flows, but shape and timing ($r_H$=0.82) and volume ($\beta_H$=0.75) could be improved. The model accurately simulates low flows volume, but has too much variability ($\gamma_L$=1.34) and is middling on shape and timing ($r_L$=0.76). The difficulty of the model to reproduce the depletion of the capacitive function may be due to the size and complexity of the catchment and the very specific influence of poljes draining over the catchment, which cannot be simulated within KarstMod platform. DE is very good (0.16, Fig. 5e). The model has negative dynamic and constant errors with a higher share of high flows, which highlight a small underestimation of the occurrence of low flows.

Both models were unable to reproduce the plateau-like behaviour observed at very high discharges (Fig. 2e), which is due to the flooding of a polje at Unica springs that influences the monitoring station. They are simulated as separate peaks, which is false in terms of model accuracy but may also have some underlying conceptual truth. Only two meteorological stations were considered, which is very few for such a large catchment (820 km$^2$). Moreover, the major recharge area (Javorniki plateau) does not have any direct climate data available. Both models have difficulties in consistently reproducing the very particular hydrological functioning of the system (influenced by polje and surface water). ANN model is more reactive, which helps for reproducing the dynamics of high floods peaks but hinders the simulation of low flows. Reservoir model has better dynamics for medium and low flows but does not always manage to reproduce high floods peaks, which may be a consequence of the simple structure of the model.

### 4.2    Source of uncertainties

Both ANN and reservoir models have similar trends on water volume and hydrological variability (Fig. 4). Overall volumes are great with $\beta$ ranging from 0.74 to 1.03. High-flow volumes are systematically underestimated with $\beta_H$ ranging from 0.71 to 1.01. Low-flow volumes are mainly overestimated – $\beta_L$ ranging from 0.99 to 1.92 – except for ANN model on Lez spring





with $\beta_L$ of 0.64. Overall hydrological variability is mainly underestimated, with only Lez and Aubach springs having $\gamma$ values slightly above 1. High-flow hydrological variability doesn't show a distinct trend, being either overestimated or underestimated depending on the studied system. Low-flow hydrological variability is mainly overestimated with $\gamma_L$ ranging from 0.95 to 2.01. These overestimations may be due to (i) improper – and generally softer – simulation of recession periods or (ii) too high sensitivity to precipitation events, especially in ANN models, inducing discharges oscillations during recession and low-flow periods. The performance on shape and timing ($r$) are mixed between the two approaches. They depend mainly on the system studied and the quality of the model, but also on the hydrological period considered.

These similar results between the two approaches highlight a common struggle to simulate extreme water conditions. As ANN and reservoir modelling approaches are very different, explanation must be sought in common factors to both approaches such as input data, observed data, internal/external system dynamics or the consideration of extreme events during calibration:

- Input data: Generally, in one-dimensional modelling approaches, input data only comes from at most few meteorological stations and does not accurately reflect the heterogeneity of meteorological processes on a catchment. Spatial variability of precipitation can be very high and not fully captured by meteorological stations, (i) resulting in different travel time and generating a different response at the spring (Ollivier et al., 2020), and (ii) hindering the simulation of very high flows (Pereira et al., 2014; Hohmann et al., 2020) – especially in areas where strong convective storms are frequent (Lobligeois et al., 2014). Temperature data are generally less heterogeneous than precipitation, although it can be affected by complex topography (Aalto et al., 2017). The uncertainties related to precipitation and temperature input in one-dimensional hydrological models can thus – partly – explain the difficulties to reproduce extreme events (Lobligeois et al., 2014; Huang et al., 2019; Ollivier et al., 2020; Bittner et al., 2021), especially high flows.

- Observed data: Discharge time series are generally derived from water height measured at the spring, using water level–discharge calibration curves. Numerous uncertainties are related to this determination method (Pelletier, 1988), including extrapolation errors for extreme values (Di Baldassarre and Montanari, 2009; Moges et al., 2021). Extreme events occur more rarely and are harder to measure, especially high flows. This can result in inaccurately observed discharge time series that are difficult to reproduce with simulations (e.g. Unica springs at very high flows).

- Internal/external system dynamics: Karst systems are inherently complex media. Internal dynamics are not necessarily captured in hydrological models (Sidle, 2006, 2021; Hartmann et al., 2017) and can be related to numerous processes in karst media, e.g. the saturation state of the system, surface water exchanges, temporary storage of water, incoming or outgoing flows from/to another aquifer, change of physical properties beyond a certain level, or karst features such as poljes or sinkholes. These complex processes do not occur systematically and can change from year to year (Ollivier et al., 2020). This can lead to difficulties in training ANN models or in adapting the structure of reservoir models.





- •   Extreme events during calibration: ANN and reservoir models are both trained on a calibration period. By definition, extreme events are rare. Therefore, models may have less opportunities to successfully fit model parameters to such events (Seibert, 2003), preferring more balanced parameters that are appropriate to the rest – and most – of the time series (Onyutha, 2019). In addition, models are generally calibrated over the whole time series

using one performance criterion against observed data. In this case, extreme events are not explicitly emphasised in the objective function. A solution could be to give more weight to the reproduction of certain parts of the time series, such as flood and dry periods (Singh and Bárdossy, 2012).

Both approaches can also benefit from a careful assessment of the calibration period. For example, the ANN model is thought to overestimate low flows in Unica springs by trying to fit the plateaus at very high discharges. In Lez spring, the

reservoir model simulates a slower draining in the aquifer (piezometric level) because it does not account for a potential change in underground dynamics. These limitations emphasise the need for a meticulous investigation of the results in regard to the characteristics of the system and the input data. Such errors can be avoided or lessened by excluding abnormal periods during the calibration, which can be justified by inaccurate input data or limitation in the conceptual model.

## 4.3    Comparison of general model properties

The main findings of this study are presented in Table 5. The extensive analysis of high and low flows did not show a clear trend, but did reveal slight differences between the two approaches for this study. For high-flow periods, results slightly favour the ANN approach (except for Qachqouch spring), with consistently accurate volumes and shape and timing (Fig. 4). ANN models also tend to achieve higher flows than reservoir models (Fig. 2). For low-flow periods, results slightly favour the reservoir approach (except for Aubach spring), with very good estimation of volumes and only a slight overestimation of

the hydrological variability (Fig. 4). The water level (Lez spring) was correctly simulated by both approaches, with only some imprecision during dry periods (Fig. 3).





**Table 5: Advantages and drawbacks of ANN and reservoir models, based on the results of this research.**

|  | ANN models | Reservoir models |
|---|---|---|
| Advantages | - Fast and reliable | - The simulation is supported by a conceptual model |
|  | - Flexible regarding input data | - Slightly better on low flows |
|  | - Slightly better on high flows | - Can be used to gain knowledge about system functioning |
|  | - Can be used to gain knowledge about input data, catchment delineation, recharge processes | - Can work with short observed time series |
| Drawbacks | - Struggle to reproduce extreme events | - Struggle to reproduce extreme events |
|  | - Need medium/long observed time series for a proper learning | - Input data generally need preprocessing |
|  | - Essentially a black-box approach | - Can be time consuming |
|  |  | - Potential platform/coding limitations |

ANN models are flexible and provide numerous advantages over reservoir models with respect to input data. It can easily integrate meteorological processes (e.g. snow dynamics, evapotranspiration) without any preprocessing of the raw data, whereas it is generally calculated beforehand in reservoir models. It is also possible to add a large amount of raw data in ANN models and let the model select those relevant for a good simulation, which make the modelling easier and also can give insight into the input data or catchment characteristics (Wunsch et al., 2022). This helps to avoid additional

uncertainties related to (i) arbitrary decisions over the raw data (e.g. choosing precipitation from one station rather than another), (ii) interpolation (when data from several meteorological stations over a catchment are available) or (iii) preprocessing (e.g. snow routine, potential evapotranspiration). This great flexibility regarding input data makes ANN model close to a 2D or semi-distributed approach. If necessary, the transition between 1D and 2D input data are comparably easy, whereas in reservoir models this usually involves changing or adapting the tool.

Reservoir models do not need long calibration period to provide accurate and relevant simulation results. In contrast, a short time series can be detrimental for the learning of ANN model, which seems to benefit from long calibration periods (at least 5 years). We have seen that the ANN model has difficulties in simulating the flows of the Qachqouch spring, mainly because of (i) the short calibration period, and (ii) the long low water periods which are not relevant for training the model. On the other hand, the reservoir model has been able to integrate key elements (e.g. double porosity, matrix-conduit exchanges, fast

conduit transfer in wet periods) by relying on the conceptual model.

The ANN approach does not require any prior knowledge of the system and inherently considers model structure and parameters. This makes the modelling process easier and faster thus saving the operator a great amount of time. On the other hand, reservoir models require a significant investment in reading the literature, analysing expert knowledge, and doing trial





and error during model design. Moreover, the cost of a change of structure is not trivial. Depending on the modelling
platform (e.g. software, raw code), it may take more or less time – or even be impossible – to take certain elements into
account. For example, in this study, the KarstMod platform does not allow (i) different porosities to be considered in the
same reservoir, leading to difficulties in modelling the piezometric levels during dry periods for the Lez system; (ii) using
different $E_{min}$ values, which may benefit the Qachqouch model; or (iii) considering polje and surface water influence in the
Unica model.

Both ANN and reservoir models can be used for research purpose. Model structure, transfer functions and parameters are
explicitly expressed in reservoir models, which can provide valuable insights into the hydrogeological structure of the
reservoir and the internal processes of the karst system, e.g. (i) the relative contributions of fast and slow flows; (ii) the
draining of each compartment; (iii) the activation thresholds of the overflow transfer functions (either to the spring or out of
the system); (iv) the changes in flow dynamics with respect to dry and wet periods; and (v) the exchanges between the
matrix and conduit compartments. In comparison, ANN models act rather as a "black-box", whose parameters are more
difficult to exploit and associate with the hydrological functioning of a system. However, ANN model can help to explore
input data, thus indirectly providing insights into catchment delineation or external recharge processes (Wunsch et al., 2022).

## 5    Conclusion

Our objective was to provide researchers and stakeholders with guidelines for choosing either artificial neural networks or
reservoir models to simulate karst spring discharges, depending on their purpose, data availability, data length and time
budget. Five test sites were considered, allowing different hydrological conditions and input data to be studied. The results of
ANN and reservoir models were compared on the basis of several performance criteria, distinguishing between high- and
low-flow conditions. Both models succeeded in simulating spring discharges satisfactorily, although struggling to reproduce
extreme events (drought, flood), generally overestimating low flows and underestimating high flows. This can be related to
common problems in hydrological modelling regarding uncertainties in the input data or observed data, internal/external
system dynamics or the consideration of extreme events during calibration.

ANN models seem robust for reproducing high-flow conditions and reservoir models for reproducing low-flow conditions.
The input data are also a critical factor of choice. Reservoir models can work with relatively short time series while ANN
models need a minimum number of relevant years to learn the functioning of a karst system. On the other hand, ANN
models are very flexible on the format and amount of input data. It can learn many meteorological processes with no prior
need for preprocessing the raw data, as well as use several time series for a single variable. This avoids arbitrary
interpolation decisions (e.g. precipitation between several stations), parameter definitions (e.g. snow routine) or
meteorological calculation (e.g. potential evapotranspiration), and allows these aspects to be integrated into the model
calibration.



Both ANN and reservoir models can be used for karst aquifer management, flood forecasting and system characterisation. ANN models may be more appropriate for simulating high flows, delineating catchments, or assessing external recharge processes. Reservoir models seems more robust for simulating low flows and gaining insights into the internal functioning of a system. ANN models can also be interesting time-wise as (i) they do not require any prior knowledge of the system and (ii) model design is more flexible regarding input data and system functioning. Given appropriate input data, both models are

suited for climate change predictions, as they showed accurate simulations with long calibration periods. However, the operator should keep in mind that the models (i) have difficulty in reproducing extreme events, which are likely to occur more frequently, and (ii) can only reproduce what they know or have learned during the calibration period and cannot take into account the impact of the evolution of recharge processes on the aquifer functioning.

One of the difficulties this paper faced was to distinguish the general limitations of the reservoir modelling approach from

those specific to the chosen modelling platform. In comparison to user-defined models, the modelling platform constrains the structure and the transfer functions of the conceptual model. Remaining within the KarstMod platform provided the time advantages of a turnkey toolbox (which are widely used in research and by stakeholder), but limited the possibilities of the conceptual models. For example, model performance could have been improved by considering the evolution of system properties with depth, or snow accumulation and melting could have been more accurate by including its parameters in the

model calibration. Implementation of a meteorological module in KarstMod for directly calibrating the parameters of the snow routine is besides under implementation.





## Appendix A: Location of the study sites



**Figure A1: Location of the study sites (carbonate outcrops from Chen et al. (2017)).**



## Appendix B: Origin of the meteorological data

**Table B1: Origin of the meteorological data (i) P, (ii) T, (iii) R$_{SO}$, (iv) RH, (v) U, (vi) AET, (vii) R$_S$, (viii) S and (ix) NS refer to (i) Precipitation, (ii) Temperature, (iii) Clear-sky solar radiation, (iv) Relative humidity, (v) Wind speed, (vi) Actual evapotranspiration, (vii) Solar radiation, (viii) Snow and (ix) New snow, respectively.**

| Spring | Station | Altitude [m] | Latitude [°] | Longitude [°] | Data measured |
|---|---|---|---|---|---|
| Aubach | Diedamskopf | 1790 | 47.3389 | 10.0256 | P, T, R$_{SO}$ |
| | Oberstdorf | 806 | 47.3984 | 10.2759 | P, T, R$_{SO}$ |
| | Walmendinger Horn | 1650 | 47.3219 | 10.1225 | P, T, R$_{SO}$ |
| Gato Cave | Grazalema | 901 | 36.7678 | -5.3658 | P, T |
| Lez | Prades-le-Lez | 69 | 43.7176 | 3.8573 | P, T, RH, U |
| | Puéchabon | 250 | 43.7414 | 3.5958 | AET |
| | Saint-Martin-de-Londres | 214 | 43.7903 | 3.7326 | P |
| | Sauteyrargues | 150 | 43.8345 | 3.9207 | P |
| | Valflaunès | 155 | 43.8001 | 3.8707 | P |
| Qachqouch | *950 m station* | 950 | 33.9180 | 35.6763 | P, T, RH, U, R$_S$ |
| | *1700 m station* | 1700 | 34.0253 | 35.8360 | P, T, RH, U, R$_S$ |
| Unica | Cerknica | 569 | 45.7956 | 14.3634 | P, S, NS |
| | Postojna | 533 | 45.7661 | 14.1932 | P, T, RH, S, NS |

## Appendix C: Calculation details for the Thiessen's polygons interpolation method

The Thiessen's polygons interpolation method consists of calculating a weighted average of precipitation data from several meteorological stations. The contribution percentages of the stations are proportional to their influence area on the catchment. An influence area corresponds to a polygon where the precipitation is considered to be identical to that measured at the associated meteorological station. The polygons are defined in two steps: (i) drawing the straight-line segments between all adjacent stations and (ii) adding the perpendicular bisectors of each segment, which correspond to the edges of the polygons. The weighted average of the precipitation $P_{TH}$ is calculated with the following equation:

$$P_{TH} = \frac{\sum_{i=1}^{n} A_i P_i}{A}$$ (C1)






With $A$ the area of the catchment, $n$ the number of meteorological stations, $A_i$ the area of the polygon associated to the $i^{th}$ station and $P_i$ the precipitation measured at the $i^{th}$ station.

**Appendix D: Calculation details for the snow routine**

Accounting for snow accumulation and melting in hydrological modelling can greatly improve model results, especially for regions with high snow volumes. Chen et al. (2018) successfully simulated spring discharge of a mountainous karst system strongly influenced by snow accumulation and melting. They applied a modified version of the HBV snow routine Bergström (1992) proposed by Hock (1999). We used this snow routine as an external KarstMod module (i.e. without internal calibration).

The snow routine simulates snow accumulation and melting over different sub-catchments defined according to altitude ranges. The input data consist in three time series (temperature, precipitation and potential clear-sky solar radiation) and five parameters (temperature threshold, melt coefficient, refreezing coefficient, radiation coefficient and water holding capacity of snow). The potential clear sky solar radiation time series and radiation coefficient are only used when working at an hourly time scale to simulate a more refined snowmelt by considering sun exposure. We calibrated the temperature
threshold, the melt coefficient and the radiation coefficient manually.

Precipitation is considered as snow when the air temperature is below the temperature threshold. Snowmelt begins when the temperature is above the threshold according to a degree-day expression, where snowmelt is a function of the melt coefficient, solar radiation and degrees above the threshold. Runoff starts when the liquid water holding capacity of snow is exceeded. The refreezing coefficient allows to consider the refreezing processes of liquid water in the snow if snowmelt is
interrupted (Bergström, 1992). The output of the snow routine is a time series of redistributed precipitation.

**Appendix E: Examples of wave-like behaviour produced by the ANN model**

The periods were selected in such a way that the influence of snow precipitation and melt is zero or almost zero. Precipitation input correspond to either direct observations from a meteorological station, or preprocessed observations with Thiessen's polygon interpolation (Appendix C) if there are several meteorological stations.







**Figure E1: Examples of wave-like behaviour produced by the ANN model on (a) Aubach, (b) Gato Cave, (c) Qachqouch and (d) Unica springs.**




**Code and data availability**

We provide complete codes for ANN models and *.properties* files for reservoir models on Github (Cinkus and Wunsch, 2022). Due to redistribution restrictions from several parties, a dataset cannot be provided. However, the data are available from the local authorities upon request. Aubach spring discharge time series and meteorological data from Diedamskopf and Walmendinger Horn stations are available from the office of the federal state of Vorarlberg – division of water management. Meteorological data from Oberstdorf station are available on the DWD Open Data Server (DWD, 2022). Gato Cave spring

discharge time series is available from the Confederación Hidrográfica de las Cuencas Mediterráneas Andaluzas (Cuenca Mediterránea Andaluza, 2022) and meteorological data is available in "Datos a la carta" section in Consejería de Agricultura, Pesa, Agua y Desarrollo rural (Consejería de Agricultura, Pesa, Agua y Desarrollo rural, 2022). Lez spring discharge time series is available on the OSU OREME website (SNO KARST, 2019), water level time series can be requested from Montpellier Méditerranée Métropole, and meteorological data are available on request from Météo-France. Qachqouch

discharge time series and meteorological data are available on request from the Department of Geology at the American University of Beirut. Unica spring discharge time series and meteorological data are available from ARSO (Slovenian Environment Agency) (ARSO, 2021a, b).

**Author contribution**

GC, AW, NM, TL, HJ and NG conceptualised the study and designed the methodology. GC and AW developed the software

code. GC and AW performed the experiments and investigated and visualised the results. GC and ZC performed formal analysis. GC wrote the original paper draft with contributions from AW, NR, JD, JFO. All authors contributed to the interpretation of the results and review and editing of the paper draft. NM and HJ supervised the work.

**Competing interests**

The authors declare that they have no conflict of interest.

**Acknowledgements**

We thank the French Ministry of Higher Education and Research for the thesis scholarship of G. Cinkus as well as the European Commission for its support through the Partnership for Research and Innovation in the Mediterranean Area (PRIMA) program under Horizon 2020 (KARMA project, grant agreement number 01DH19022A). The data collection and instrumentation on the Qachqouch catchment were funded by USAID and National Academy of Science (Peer Science;

project award: 102881; Cycle 3) and KARMA project (L-CNRS in the framework of the PRIMA program; Award# 103895; Project# 25713). Moreover, Beirut and Mount Lebanon Water Establishment are thanked to facilitate the installation of





instruments and access to field sites. We thank the Slovenian Research Agency for financial support within the project "Infiltration processes in forested karst aquifers under changing environment" (No. J2-1743) and the Karst Research Programme (No. P6-0119). We further thank the National Agency of Research of the Spanish Ministry of Science,

Innovation and Universities for the funding of the KARMA project (PCI2019-103675) and the Autonomous Government of Andalusia (Spain) for the funding of the Research Group RNM-308. For the data provided, we also acknowledge the French Karst National Observatory Service (SNO KARST, 2019), the French national meteorological service Météo-France, the office of the federal state of Vorarlberg – division of water management, the German Meteorological Service, the Slovenian Environment Agency (ARSO, 2021a, b) and the Meteorological National Agency (AEMET).

The manuscript was written with the Rmarkdown framework (Allaire et al., 2021; Xie et al., 2018, 2020), using R (R Core Team, 2021) and the following packages: readxl, readr, dplyr, tidyr, ggplot2, lubridate (Wickham et al., 2019), cowplot (Wilke, 2020), flextable (Gohel, 2021), hydroGOF (Mauricio Zambrano-Bigiarini, 2020) and padr (Thoen, 2021).

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
