# Peer review of "Comparison of artificial neural networks and reservoir models for simulating karst spring discharge on five test sites in the Alpine and Mediterranean regions"

_Hydrology and Earth System Sciences, 2022_

## Referee Comment (RC2)

Dear Authors,

The paper is comparing ability of an Artificial neural network (ANN) model – Convolutional Neural Network model (CNN), and a reservoir model – KarstMod in five European karst springs to simulate discharges.

The idea of comparing ANNs and reservoir models in the context of karst spring discharge simulation stands as an interesting point of interest for hydrology and earth system sciences, especially through consideration of the different basins with different characteristics. Therefore, the article is considered relevant to the scope of HESS since it can contribute the improvement of karst system modelling approaches with extensive comparisons between different models. However, following suggestions and comments can be considered to improve the context and methodology of the paper and provide better understanding to readers.

1- In Data and study sites section (2.1 to 2.5) (p3. From 85-165): it could be better to explain the reason behind using different potential evapotranspiration (PTE) calculation methods for different basins and the impact of different methods on PTE, if there is any.

2- In Artificial neural networks section (3.1) (p6. From 175 to 185):
Selected ANN model type should better be demonstrated with figures as it is done for reservoir models in the following section. Since CNNs are quite complex deep learning models, it can be difficult to be comprehended even for experienced ANN modelers. Demonstrations of CNN and a little more detailed explanation of the regularization methods to avoid traps such as exploding gradient could enrich the context of the article.

3- In Model evaluation section section (3.4) Model evaluation criteria should be better justified. Moreover, using more evaluation criteria can be more rigorous which are usually considered in hydrological modelling studies. For instance, relative volume bias, normalized peak error, root mean square error, nash score (the last two is suggested sice they are also used as cost function).

4- In Introduction section (p. 2) from line 48 to 51, authors said "Distributed models require a lot of data for defining physical parameters and thus can 50 be tough to use in a scarce data context. On the other hand, data-driven models permit studying complex and heterogeneous karst systems without requiring extensive meteorological and system-related data.". However, the difference between distributed and data-driven models (if the authors meant statistical models such as ANN) about the data requirement is not necessarily about the amount of data but rather about the diversity. This phrase can be modified to indicate that distributed models need more diverse data while ANNs need only input and output data.

5- In Introduction section (1) (p. 2) from line 57 to 62 references can be given in historical order.

   In Introduction section (1) (p. 3) in line 74 references can be given in historical order.

   In Reservoir model section (3.2) (p. 7) in line 74 references can be given in historical order.

   [References can be homogenized as so…]

6- In Source of uncertainties section (4.2) (p.23) from line 518 to 540, authors can mention the expected (if not quantified) amount of uncertainty for each source in their case studies for different basins either by expertise or literature support. Since basins are of different characteristics, it can be interesting to have such information for readers.

---

## Author Comment (AC1)

We thank the reviewer for the very relevant major comments, which helped us to improve the quality and relevance of the models. To address comment 1.1 (use same performance metric for the calibration) and comment 1.5 (calibration of the parameters of the snow routine), we reran the reversoir models using KarstMod in command line, thus allowing (i) to use the same performance metric (MSE) for both modelling approaches and (ii) to calibrate the parameters of the snow routine.

**1.1 Line 290: I think the authors must explain why they used different performance metrics for the calibration of the models. Choosing the objective function(s) is the subjective decision of the modeller, however, if you want to make a comparison between the model performances, I recommend calibrating the models according to the same performance metric.**

Initially, we used different performance criteria because MSE is the most common metric for calibrating ANN models but is not implemented in KarstMod. However, we agree with the reviewer that the comparison of the results will be more relevant using the same performance metric for calibration. We used KarstMod in command line, thus allowing to use MSE as a performance metric for the reservoir models. Both ANN and reservoir models are now calibrated using the same performance metric. The figures 2, 3, 4, 5, the table 4 and the text of section 4 have been updated with the new results and also include the MSE for the evaluation of model performance.

[Figure]

[Figure]

**1.2    Line 294: Is this adequate for a successful calibration?**

Indeed, this can be improved for the reservoir modelling approach. As suggested in comment 1.3, we reran the reservoir models during a 6-hour period and selected all the simulations above a specific threshold value for the objective function (MSE).

**1.3    Line 298: This is not a scientific statement. Running the model for 1 hour does not tell anything about the number of simulations. Instead, I recommend applying a threshold value (e.g. NSE > 0.5) for the behavioural runs and selecting all of the simulation results above this threshold rather than selecting the top 1000 simulations. By doing this, you would probably get a different number of behavioural model runs in each catchment and you can better compare the model uncertainty bounds. (Alternatively, you may mention the stop criteria of the KarstMod model e.g. simulation-time limitation, or number of behavioural runs)**

Thank you for the relevant comment. We indeed used a threshold value for selecting all the simulation results but this was poorly phrased in the manuscript. We reran all the models using a maximum MSE threshold on the calibration period for a 6-hour model run and modified the manuscript accordingly, L298: *"In KarstMod (reservoir models), the retained simulations correspond to all the results satisfying a maximum MSE threshold on the calibration period for a 6-hour model run."*

**1.4    Line 305: Here you mention that you evaluated the model performances by using KGE and DE, but in line 290, you calibrated the models by using NSE (reservoir model) and MSE (ANN model) metrics. So, why did you change the objective function in the model evaluation phase? Please explain. As far as I experienced, the best simulation obtained by any of the performance metrics (e.g. NSE) does not guarantee the best simulation on another metric (eg. KGE). So, please calibrate and evaluate the models again by using the same performance metric(s).**

Thank you for the comment. As explained in comment 1.1, we calibrated and evaluated the models again using the same performance metric (MSE). We also added the MSE results for the validation period in Figure 4 and Table 4. However, we keep the in-depth evaluation of the models results with the modified KGE and its components, as it provides valuable insights into the different aspects of a model (i.e. variability, volume and correlation).

**1.5    Line 350: This plot shows that the reservoir model outperforms the ANN model in Spanish, French, Lebanese and Slovenian catchments except for the Austrian catchment. I think the Austrian catchment is dominated by snow (as you mentioned in line 379) and the reservoir model structure is not adequate to beat the ANN model. It must be discussed in the discussion part. Additionally, your snow routine may not be adequate if you did not calibrate the snow parameters. Please see the paper (Çallı et.al. 2022).**

Please see comment 1.18 about the model structure of the Aubach model. Following your suggestion (also comments 1.19 and 1.24), we reran the reservoir models including the estimation of the parameters of the snow routine by model calibration. Despite the catchments (except Aubach) not being hugely affected by snow, this helped to get more relevant results while strengthening the methodology of the reservoir modelling approach.

**1.6     Lines 39-40: In my opinion, you would have a better classification of the models. Kovacs and Sauter (2007) do not classify the models as "data-driven" or "distributed" models. Distributed or lumped, all models are somehow data-dependent. I think that would be better to classify the models as "black-box models", "conceptual models" and "physical models" considering their complexities. You may give details about the "Machine learning models" under the "Black Box", and reservoir models (or lumped parameter models) under the "Conceptual models". You may also give some details about the advantages and disadvantages of these modelling approaches regarding the complexities and data requirements. You may mention why so many researchers choose conceptual models. Another point is that, please be more consistent about the model classification inside the paper.**

We agree, the classification of modelling approaches as "distributed" or "lumped parameter" is better suited and consistent with the classification of Kovacs and Sauter (2007). We modified the manuscript accordingly, also giving details about (i) the neural networks models under the "black box" class, and (ii) the reservoir models under the "lumped parameter" class, L43: *"They include (i) "black-box" models such as neural networks-based approaches, which use no a priori information about the functioning of a system; and (ii) "conceptual" models, which are based on a conceptual representation of a karst system – e.g., for the reservoir models, a succession of one or several reservoirs using simplified physical transfer functions."* The advantages and disadvantages of these modelling approaches are extensively detailed and discussed in section 4.3 (Comparison of general model properties). We added a sentence in the introduction to explain why conceptual models such as reservoir models are well suited to the study of karst systems: *"This approach is well suited to karst systems due to the high heterogeneity and low level of knowledge of their structure (Fleury et al., 2009; Hartmann et al., 2012)."*

**1.7     Line 47: You may consider citing the paper (Addor and Melsen, 2019) about the model selection procedure (adequacy or legacy).**

Indeed, this is an important factor of choice. We modified the manuscript accordingly, L47: *"The choice of a modelling approach depends mainly on the objective of the study, but also on the current knowledge of the system, the available data and regional/institutional preferences (Addor and Melsen, 2019)."*

**1.8     Line 50: You may consider rephrasing the sentence "…distributed models require a lot of data". You may alternatively say: "Distributed models require the data with high spatial resolution, however, lumped models require data in high temporal resolution." (You may cite here again Hartmann et.al. 2014 or Kovacs and Sauter 2007).**

We thank the reviewer for the suggestion. Indeed, it is relevant to detail what type of data is needed in distributed and data-driven models. We modified the manuscript accordingly.

L49: *"Distributed models require a lot of diverse data with high spatial and temporal resolution for defining physical parameters and thus can be tough to use in a scarce data context (Hartmann et al., 2014)."*

L50: *"On the other hand, data-driven models permit studying complex and heterogeneous karst systems without requiring extensive meteorological and system-related data with high spatial resolution."*

**1.9     Line 51: I would remove the sentence "Both black-box and reservoir models …." to avoid repetition. The following sentences already explain the applications of the ANN and reservoir models for academic and operational purposes.**

As suggested, the sentence was removed.

**1.10    Lines 55-60: Please be more consistent about the references in the brackets. You may use time order (Perrin et.al. 2003; Jukic and Denic-Jukic 2009; Tritz et.al 2011; Bittner et.al. 2020) or alphabetical order (Bittner et.al. 2020; Jukic and Denic-Jukic 2009; Perrin et.al. 2003; Tritz et.al 2011).**

All the references in brackets were ordered by time order.

**1.11    Line 81: The paper does not include any simulations by using the artificial future data (different emission scenarios) to compare the model adaptability against climate change. So this is not fair to have such an inference. If you want to declare that the models are not robust in extreme event predictions, please cite several climate-related studies (to better link the connection between the climate-change and extreme events).**

We agree that it is not appropriate to provide inferences on the models' suitability for climate change predictions. We removed the sentences concerned (in the introduction and conclusion).

**1.12    Line 91: I recommend moving map A.1 to this section.**

We moved Figure A1 to section 2 (main text, now Figure 1).

**1.13    Line 174-175: In this sub-section, adding a schematic illustration of the ANN model would be very helpful to better understand the modelling architecture.**

A figure was added to explain the modelling architecture of the ANN model.

[Figure]

**1.14    Line 180: Please simply explain why you applied the 1-D convolutional layer approach (you may consider citing previous ANN modelling studies).**

A sentence was added in the introduction to explain why we apply 1D Convolutional Neural Network: *"We specifically apply 1D Convolutional Neural Networks (CNNs) because in an earlier study (Wunsch et al., 2022) we were able to demonstrate their high ability to perform karst spring discharge modelling. Furthermore, they have some favourable properties compared to popular recurrent neural networks (e.g. the LSTMs), such as batch-wise training procedure which makes them considerably faster and computationally less expensive."*

**1.15    Line 184: I think you can move the names of the python libraries to the appendix. It is not necessary inside the text.**

The names of the Python libraries were moved in the *Acknowledgements* section.

**1.16    Line 185: Here I see you used the library BayesOpt. Was that library used for the Bayesian uncertainty analysis? Or did you apply an uncertainty analysis to the ANN model predictions? If so, please explain which method you applied. Please give some details about the model assumptions, parameter distributions etc.**

Actually, we used the library Bayesian Optimization, not BayesOpt. We apologise for this inaccuracy. We did not perform a Bayesian uncertainty analysis with this package, but used it to optimise the ANN hyperparameters. This optimisation strategy unfortunately was not yet explained in the paper. We therefore added clarifying statements and explanations and thank the reviewer for noticing. Please note that the uncertainty is estimated from the model ensemble with 1000 members, which in the case of the CNN are generated by a combination of different model initializations and runs based on monte carlo dropout. A paragraph was added: *"Besides number of filters and number of neurons in the first dense layers, we optimised the training batch size and the length of the input sequence for each simulation step using the Bayesian Optimization library (Nogueira, 2014). The number of minimum and maximum optimisation steps was individually selected for each site and can be found in the provided modelling scripts (Cinkus and Wunsch, 2022)."* The reference of the Python library was corrected, L183: *"[...] Bayesian Optimization (Nogueira, 2014) [...]"*

**1.17    Line 189: You may consider pointing out the functionality of the conceptual models in karst water predictions. What is the main advantage of this modelling approach?**

Please see the response of comment 1.6. The main advantage of the conceptual models regarding karst water predictions is now detailed in the introduction.

**1.18    Line 240: You may consider adding a snow reservoir above the Epikarst in Fig 1a. This would be more suitable for the mountainous catchment.**

Indeed, it would be more appropriate. However, this option is not available in the KarstMod platform, and appears to be one of the limitations of using a platform for reservoir modelling (similar to the consideration of polje and surface water influence in the Unica model). We added some details about this aspect both in the Aubach and Discussion sections:

L378: *"These errors can be either due to (i) a miscalibration of the snow routine, retaining too much water as snow in winter and thus releasing too much in warmer periods, or (ii) the snow dynamics which cannot be taken into account within the KarstMod platform, e.g. by adding a snow storage above the epikarst (Chen et al., 2018)."*

L588: *"[...] which may benefit the Qachqouch model; (iii) considering polje and surface water influence in the Unica model; or (iv) considering snow dynamics in the structure of the Aubach model."*

**1.19    Line 261: Please mention how to determine the snow routine parameters (Degree-day factor and melting temperature). You mentioned that you did not make an optimization for the snow parameters, so please cite the relevant literature (e.g. He et.al. 2014).**

The parameters of the snow routine are now estimated by model calibration. Please see the response to comment 1.5 for more details.

**1.20    Line 343: The uncertainty bounds are not easy to see especially for the reservoir model in (a). I think when you select all the behavioural simulations (above the threshold), the uncertainty bound will be much more visible. Then the reader can make a visual comparison between them.**

This has been improved with the rerun of the models. Please see the responses of comments 1.1 and 1.3 for more details.

**1.21    Line 347: Again the same problem. Please apply a threshold for the reservoir model, and use all the behavioural runs to obtain the uncertainty bounds.**

This has been improved with the rerun of the models. Please see the responses of comments 1.1 and 1.3 for more details.

[Figure]

**1.22    Line 355: Please share the model calibration skills in the table.**

You are right that the information is interesting to provide for reservoir models. However, the score of an ANN model during the calibration period is not relevant as it corresponds to its learning period. Thus, it seems more appropriate to add this detail in appendix to avoid potential confusion for the reader. We added a table that shows the calibration and validation scores of the reservoir models (with a comparison to the ANN models for the validation period).

**1.23    Line 355: You mentioned that the ANN models require long time series to learn the functionality of the karst system. On the other hand, reservoir models could be calibrated for relatively shorter periods. But, there are some results to be discussed in detail                                              as                                              below:**

**Lez and Qachqouch springs simulation results support the hypothesis, but the Aubach catchment does not. We expect better calibration skills in the reservoir model in a short calibration period. However, the ANN model outperforms the reservoir model in Aubach. How do you explain this?**

While Aubach and Qachqouch springs have similar lengths for the calibration period (nearly six years, and about four years, respectively), Lez spring has a longer calibration period of slightly more than nine years, which we consider not exactly a "short" period. For Qachqouch, it is explained Line 511 that "even when data are available, there is a significant amount of time without (relevant) discharge, for which no input-output relation can be learned." In comparison, Aubach is a very reactive system with a lot of "relevant" discharge, which benefit the training of the ANN model. The difference between the springs' regime of Aubach and Qachqouch can be appreciated in Figure 2. We modified the manuscript to emphasise the aspect of relevant discharge in short time series:

L473: *"This highlights the strength of conceptual modelling to take into account recharge processes and reservoir replenishment, even on a short dataset with long dry periods."*

L575: *"In contrast, a very short time series or a short time series with long dry periods can be detrimental for the learning of ANN model, which seems to benefit from medium/long periods of relevant discharge (at least 5 years)."*

L25: *"[...] (ii) reservoir models can provide good results even with few years of relevant discharge in the calibration period, [...]"*

**1.24    Line 389: You can make a representative snow routing even if you do not calibrate the snow parameters. Please cite the relevant literature.**

The parameters of the snow routine are now estimated by model calibration. Please see the response to comment 1.5 for more details.

**1.25    Line 527: You may discuss the uncertainty in the temperature data. Temp data strongly affect the timing of the recharge, especially in snow-covered areas (Aubach case).**

A sentence was added to discuss the uncertainties related to temperature data in snow-covered areas, L524: *"In the case of snow-covered areas, it can result in strong uncertainties on the timing of snow accumulation and melting (Zhang et al., 2016), and therefore the recharge of the aquifer."*

**1.26    Line 547: You may add some other model optimization techniques (e.g. cross-validation, see Wilks 2011).**

We agree with the suggestion and modified the manuscript accordingly, L546: *"[...] or to use different model optimisation techniques, such as cross-validation (Wilks, 2011)."*

**1.27    Line 559: Please discuss the model's structural adequacy here.**

Several sentences were added to discuss the model's structural adequacy:

L556: *"For high-flow periods, results slightly favour the ANN approach (except for Qachqouch spring), with consistently accurate volumes and shape and timing (Fig. 6). ANN models also tend to achieve higher flows than reservoir models (Fig. 4); due to the noticeable/strong karstification of the studied systems, the high occurrence of high discharge data may benefit the learning of the ANN models. On the other hand, reservoir models are more dependent on the relevance and the quality of the input data preprocessing, thus can be more affected by the uncertainties presented in Sect. 4.2, especially regarding high flows."*

L558: *"For low-flow periods, results slightly favour the reservoir approach (except for Aubach spring), with very good estimation of volumes and only a slight overestimation of the hydrological variability (Fig. 6). The conceptual representation of the aquifer with reservoirs and transfer functions may help to simulate the recharge process (especially for inertial systems): a precipitation event will not directly result in a discharge increase at the spring if the reservoir is not fully saturated. On the other hand, ANN models seem to not always account for the time needed for the aquifer to replenish, inducing wave-like behaviours during medium- and low-flow periods (Fig. D1), which can hinder the simulation of low flows."*

**References**

Addor, N. and Melsen, L. A.: Legacy, Rather Than Adequacy, Drives the Selection of Hydrological Models, Water Resources Research, 55, 378–390, https://doi.org/10.1029/2018WR022958, 2019.

Wilks, D. S.: Statistical Forecasting, in: International Geophysics, vol. 100, Elsevier, 215–300, https://doi.org/10.1016/B978-0-12-385022-5.00007-5, 2011.

Zhang, J. L., Li, Y. P., Huang, G. H., Wang, C. X., and Cheng, G. H.: Evaluation of Uncertainties in Input Data and Parameters of a Hydrological Model Using a Bayesian Framework: A Case Study of a Snowmelt–Precipitation-Driven Watershed, Journal of Hydrometeorology, 17, 2333–2350, https://doi.org/10.1175/JHM-D-15-0236.1, 2016.

---

## Author Comment (AC2)

We thank the referee for their careful reading and helpful comments. Our reply is given below.

**2.1      In Data and study sites section (2.1 to 2.5) (p3. From 85-165): it could be better to explain the reason behind using different potential evapotranspiration (PTE) calculation methods for different basins and the impact of different methods on PTE, if there is any.**

We agree that it is better to explain why different methods are being used for calculating potential evapotranspiration. We think that there is no negative impact of using different methods; on the contrary, the methods are carefully chosen to provide the most relevant estimations, taking into account the available meteorological data, the climate of the area and local expert knowledge. The manuscript was modified accordingly, L261: *"For reservoir models, evapotranspiration processes were considered using time series of potential evapotranspiration, which were calculated for each site using different methods depending on the available meteorological data, the climate of the area and local expert knowledge."*

**2.2      In Artificial neural networks section (3.1) (p6. From 175 to 185): Selected ANN model type should better be demonstrated with figures as it is done for reservoir models in the following section. Since CNNs are quite complex deep learning models, it can be difficult to be comprehended even for experienced ANN modelers. Demonstrations of CNN and a little more detailed explanation of the regularization methods to avoid traps such as exploding gradient could enrich the context of the article.**

A figure has been added to better explain the functioning of ANN models. Further details have also been added to the description of the ANN modelling approach in section 3.1: *"To ensure proper learning, the models are regularised with several measures. Hence, early stopping with a patience of 20 Steps is applied to prevent the model from overfitting. Except Qachqouch, where little data is available, the size of the according stopset ranges between one and four annual cycles (see provided scripts for details). This stopset is considered a part of the calibration period mentioned in Sect. 3.4. Further, dropout ensures robust training and serves as another measure against overfitting. We applied the Adam optimizer for a maximum of 150 to 300 training epochs with an initial learning rate of 0.001 and applied gradient clipping to prevent exploding gradients."*

[Figure]

**2.3      In Model evaluation section section (3.4) Model evaluation criteria should be better justified. Moreover, using more evaluation criteria can be more rigorous which are usually considered in hydrological modelling studies. For instance, relative volume bias, normalized peak error, root mean square error, nash score (the last two is suggested sice they are also used as cost function).**

Thanks for the recommendation, we added the MSE to the list of evaluation criteria as it is the performance metric used for calibrating the ANN and reservoir models. Now the models are evaluated using MSE, modified KGE, Diagnostic Efficiency, $\beta$ for the volume bias, $\gamma$ for the discharge variability and the Pearson correlation coefficient $r$ for the discharge shape and timing – Figure 4 was updated (see below). We also explained in more details why we evaluate the results using the modified KGE instead of the Nash-Sutcliffe Efficiency: *"The Kling-Gupta Efficiency (KGE) has gained in popularity as it aims to address some limitations of the Nash-Sutcliffe Efficiency (Nash and Sutcliffe, 1970), i.e. (i) the discharge variability is underestimated, (ii) the mean of observed values is not a meaningful benchmark for variables with high variability, and (iii) the normalised bias is dependent to the variability (Gupta et al., 2009, Willmott et al., 2012)."*

[Figure]

**2.4 In Introduction section (p. 2) from line 48 to 51, authors said "Distributed models require a lot of data for defining physical parameters and thus can 50 be tough to use in a scarce data context. On the other hand, data-driver models permit studying complex and heterogeneous karst systems without requiring extensive meteorological and system-related data.". However, the difference between distributed and data-driven models (if the authors meant statistical models such as ANN) about the data requirement is not necessarily about the amount of data but rather about the diversity. This phrase can be modified to indicate that distributed models need more diverse data while ANNs need only input and output data.**

We thank the reviewer for the suggestion. The sentence has been modified to emphasise that distributed models need more diverse data (both with high spatial and temporal resolution), in contrast to lumped parameter models:

L49: *"Distributed models require a lot of diverse data with high spatial and temporal resolution for defining physical parameters and thus can be tough to use in a scarce data context (Hartmann et al., 2014)."*

L50: *"On the other hand, data-driven models permit studying complex and heterogeneous karst systems without requiring extensive meteorological and system-related data with high spatial resolution."*

**2.5 In Introduction section (1) (p. 2) from line 57 to 62 references can be given in historical order. In Introduction section (1) (p. 3) in line 74 references can be given in historical order. In Reservoir model section (3.2) (p. 7) in line 74 references can be given in historical order. [References can be homogenized as so...]**

All the references were ordered in historical order.

**2.6 In Source of uncertainties section (4.2) (p.23) from line 518 to 540, authors can mention the expected (if not quantified) amount of uncertainty for each source in their case studies for different basins either by expertise or literature support. Since basins are of different characteristics, it can be interesting to have such information for readers.**

Such information is unfortunately not available for the basins of this study, either by expertise or literature support. However, in the literature, several authors focused their work on quantifying the uncertainties related to input and observed data in the context of karst aquifers. We have added details about this for each source of uncertainty in section 4.2:

Input data: *"McMillan et al. (2018) suggested that uncertainties in precipitation data are about 0–10 % at point scale but can go up to 40 % when considering interpolation uncertainties."*

Observed data: *"The uncertainties related to discharge measurements are highly dependent on the quality of the gauging station and usually range between 10–40 % (McMillan et al., 2018). Although they are expected to be higher in a karst context (Westerberg et al., 2016), some authors reported uncertainties of about 20 % (Jeannin et al., 2021) or 10–15 % (Katz et al., 2009)."*

To the best of our knowledge, quantitative information about model structure uncertainties in karst environments has never been explicitly mentioned.

**References**

Jeannin, P.-Y., Artigue, G., Butscher, C., Chang, Y., Charlier, J.-B., Duran, L., Gill, L., Hartmann, A., Johannet, A., Jourde, H., Kavousi, A., Liesch, T., Liu, Y., Lüthi, M., Malard, A., Mazzilli, N., Pardo-Igúzquiza, E., Thiéry, D., Reimann, T., Schuler, P., Wöhling, T., and Wunsch, A.: Karst modelling challenge 1: Results of hydrological modelling, J. Hydrol., 600, 126508, https://doi.org/10.1016/j.jhydrol.2021.126508, 2021.

Katz, B. G., Sepulveda, A. A., and Verdi, R. J.: Estimating Nitrogen Loading to Ground Water and Assessing Vulnerability to Nitrate Contamination in a Large Karstic Springs Basin, Florida1, JAWRA Journal of the American Water Resources Association, 45, 607–627, https://doi.org/10.1111/j.1752-1688.2009.00309.x, 2009.

McMillan, H. K., Westerberg, I. K., and Krueger, T.: Hydrological data uncertainty and its implications, WIREs Water, 5, e1319, https://doi.org/10.1002/wat2.1319, 2018.

Nash, J. E. and Sutcliffe, J.: River flow forecasting through conceptual models: Part 1. A discussion of principles., J. Hydrol., 10, 282–290, 1970.

Westerberg, I. K., Wagener, T., Coxon, G., McMillan, H. K., Castellarin, A., Montanari, A., and Freer, J.: Uncertainty in hydrological signatures for gauged and ungauged catchments, Water Resources Research, 52, 1847–1865, https://doi.org/10.1002/2015WR017635, 2016.

Willmott, C. J., Robeson, S. M., and Matsuura, K.: A refined index of model performance, Intern. J. Climatol., 32, 2088–2094, https://doi.org/10.1002/joc.2419, 2012.

---

## Author Response (AR1)

**Manuscript hess-2022-365 – Responses to Reviewers**

We thank the referees for their careful reading and helpful comments. Our reply is given below. The page and line numbers (in the "modification to manuscript" sections) correspond to the modifications done on the revised manuscript with changes marked.

**Reviewer 1**

We thank the reviewer for the very relevant major comments, which helped us to improve the quality and relevance of the models. To address comment 1.1 (use same performance metric for the calibration) and comment 1.5 (calibration of the parameters of the snow routine), we reran the reversoir models using KarstMod in command line, thus allowing (i) to use the same performance metric for both modelling approaches (MSE) and (ii) to calibrate the parameters of the snow routine.

**1.1    Quoting:** "Line 290: I think the authors must explain why they used different performance metrics for the calibration of the models. Choosing the objective function(s) is the subjective decision of the modeller, however, if you want to make a comparison between the model performances, I recommend calibrating the models according to the same performance metric."

> **Response.** Initially, we used different performance criteria because MSE is the most common metric for calibrating ANN models but is not implemented in KarstMod. However, we agree with the reviewer that the comparison of the results will be more relevant using the same performance metric for calibration. We used KarstMod in command line, thus allowing to use MSE as a performance metric for the reservoir models. Both ANN and reservoir models are now calibrated using the same performance metric.
>
> **Modification to manuscript.**
>
> - **Table 3.** The column "Objective function" now only shows "MSE".
> - **Page 13. Line 315.** The sentence was changed into: "We calibrated the models by applying the Mean Squared Error (MSE) on simulated and observed discharge time series."
> - **Page 14. Line 330.** The sentence was changed into: "We evaluated the performance of the models using the MSE and several performance criteria [...]."
> - **Page 14. Line 333.** The sentence was changed into: "For Lez spring, we also applied the MSE and KGE' criteria on water level."
> - **Figure 4, Figure 5, Figure 6, Figure 7, Table 4.** Updated with the new results from the rerun of the reservoir models. Results now also include the MSE for the evaluation of model performance.
> - **Pages 15–32. Section 4.** Text updated with the new results from the rerun of the reservoir models.

**1.2    Quoting:** "Line 294: Is this adequate for a successful calibration?"

> **Response.** Indeed, this can be improved for the reservoir modelling approach. As suggested in comment 1.2 and 1.3, we reran the reservoir models during a 6-hour period and selected all the simulations above a specific threshold value for the objective function (MSE). Please see comment 1.3 for the modifications done to the manuscript.

**1.3    Quoting:** "Line 298: This is not a scientific statement. Running the model for 1 hour does not tell anything about the number of simulations. Instead, I recommend applying a threshold value (e.g. NSE > 0.5) for the behavioural runs and selecting all of the simulation results above this threshold rather than selecting the top 1000 simulations. By doing this, you would probably get a different number of behavioural model runs in each catchment and you can better compare the model uncertainty bounds. (Alternatively, you may mention the stop criteria of the KarstMod model e.g. simulation-time limitation, or number of behavioural runs)"

> **Response.** Thank you for the relevant comment. We indeed used a threshold value for selecting all the simulation results but this was poorly phrased in the manuscript. We reran all the models using a maximum MSE threshold on the calibration period for a 6-hour model run.
>
> **Modification to manuscript.**
>
> - **Page 13. Line 319.** The sentence was changed into: "Multiple simulations were carried out for each modelling approach at each site."
> - **Page 13. Line 323.** The sentence was changed into: "In KarstMod (reservoir models), the retained simulations correspond to all the results satisfying a maximum MSE threshold on the calibration period for a 6-hour model run."

**1.4    Quoting:** "Line 305: Here you mention that you evaluated the model performances by using KGE and DE, but in line 290, you calibrated the models by using NSE (reservoir model) and MSE (ANN model) metrics. So, why did you change the objective function in the model evaluation phase? Please explain. As far as I experienced, the best simulation obtained by any of the performance metrics (e.g. NSE) does not guarantee the best simulation on another metric (eg. KGE). So, please calibrate and evaluate the models again by using the same performance metric(s)."

> **Response.** Thank you for the comment. As explained in comment 1.1, we calibrated and evaluated the models again using the same performance metric (MSE). We also added the MSE results for the validation period in Figure 6 and Table 4. However, we keep the in-depth evaluation of the models results with the modified KGE and its components, as it provides valuable insights into the different aspects of a model (i.e. variability, volume and correlation). Please see comment 1.1 for the modifications done to the manuscript.

**1.5    Quoting:** "Line 350: This plot shows that the reservoir model outperforms the ANN model in Spanish, French, Lebanese and Slovenian catchments except for the Austrian catchment. I think the Austrian catchment is dominated by snow (as you mentioned in line 379) and the reservoir model structure is not adequate to beat the ANN model. It must be discussed in the discussion part. Additionally, your snow routine may not be adequate if you did not calibrate the snow parameters. Please see the paper (ÇallÄ± et.al. 2022)."

> **Response.** Please see comment 1.18 about the model structure of the Aubach model. Following your suggestion (also comments 1.19 and 1.24), we reran the reservoir models including the estimation of the parameters of the snow routine by model calibration. Despite the catchments (except Aubach) not being hugely affected by snow, this helped to get more relevant results while strengthening the methodology of the reservoir modelling approach.
>
> **Modification to manuscript.**

- **Appendix C. Line 723.** The sentence was changed into: "The parameters values were estimated by model calibration."
- **Page 33. Line 687.** The sentence was removed: "For example, model performance could have been improved by considering the evolution of system properties with depth, or snow accumulation and melting could have been more accurate by including its parameters in the model calibration."

**1.6   Quoting:** "Lines 39-40: In my opinion, you would have a better classification of the models. Kovacs and Sauter (2007) do not classify the models as "data-driven" or "distributed" models. Distributed or lumped, all models are somehow data-dependent. I think that would be better to classify the models as "black-box models", "conceptual models" and "physical models" considering their complexities. You may give details about the "Machine learning models" under the "Black Box", and reservoir models (or lumped parameter models) under the "Conceptual models". You may also give some details about the advantages and disadvantages of these modelling approaches regarding the complexities and data requirements. You may mention why so many researchers choose conceptual models. Another point is that, please be more consistent about the model classification inside the paper."

> **Response.** We agree, the classification of modelling approaches as "distributed" or "lumped parameter" is better suited and consistent with the classification of Kovacs and Sauter (2007). We modified the manuscript accordingly, also giving details about (i) the neural networks models under the "black box" class, and (ii) the reservoir models under the "lumped parameter" class. The advantages and disadvantages of these modelling approaches are extensively detailed and discussed in section 4.3 (Comparison of general model properties). We added a sentence in the introduction to explain why conceptual models such as reservoir models are well suited to the study of karst systems.
>
> **Modification to manuscript.**
>
> - "data-driven" was replaced with "lumped parameter".
> - **Page 2. Line 43.** The sentence was changed into: "They include (i) "black-box" models such as neural networks-based approaches, which use no a priori information about the functioning of a system; and (ii) "conceptual" models, which are based on a conceptual representation of a karst system – e.g., for the reservoir models, a succession of one or several reservoirs using simplified physical transfer functions."
> - **Page 2. Line 60.** The sentence was moved from section 3.2 (Reservoir models) to section 1 (Introduction): "This approach is well suited to karst systems due to the high heterogeneity and low level of knowledge of their structure (Fleury et al., 2009; Hartmann et al., 2012)."

**1.7   Quoting:** "Line 47: You may consider citing the paper (Addor and Melsen, 2019) about the model selection procedure (adequacy or legacy)."

> **Response.** Indeed, this is an important factor of choice. We modified the manuscript accordingly.
>
> **Modification to manuscript. Page 2. Line 47.** The sentence was changed into: "The choice of a modelling approach depends mainly on the objective of the study, but also on the current knowledge of the system, the available data and regional/institutional preferences (Addor and Melsen, 2019)."

**1.8   Quoting:** "Line 50: You may consider rephrasing the sentence "…distributed models require a lot of data". You may alternatively say: "Distributed models require the data with high spatial resolution, however, lumped models require data in high temporal resolution." (You may cite here again Hartmann et.al. 2014 or Kovacs and Sauter 2007)."

**Response.** We thank the reviewer for the suggestion. Indeed, it is relevant to detail what type of data is needed in distributed and data-driven models. We modified the manuscript accordingly.

**Modification to manuscript.**

- **Page 2. Line 49.** The sentence was changed into: "Distributed models require a lot of diverse data with high spatial and temporal resolution for defining physical parameters and thus can be tough to use in a scarce data context (Hartmann et al., 2014)."
- **Page 2. Line 51.** The sentence was changed into: "On the other hand, data-driven models permit studying complex and heterogeneous karst systems without requiring extensive meteorological and system-related data with high spatial resolution."

**1.9   Quoting:** "Line 51: I would remove the sentence "Both black-box and reservoir models …." to avoid repetition. The following sentences already explain the applications of the ANN and reservoir models for academic and operational purposes."

**Modification to manuscript. Page 2. Line 53.** As suggested by the reviewer, the following sentence was removed: "Both "black-box" and reservoir models are therefore relevant for operational and research applications".

**1.10   Quoting:** "Lines 55-60: Please be more consistent about the references in the brackets. You may use time order (Perrin et.al. 2003; Jukic and Denic-Jukic 2009; Tritz et.al 2011; Bittner et.al. 2020) or alphabetical order (Bittner et.al. 2020; Jukic and Denic-Jukic 2009; Perrin et.al. 2003; Tritz et.al 2011)."

**Modification to manuscript.** All the references in brackets were ordered by time order.

**1.11   Quoting:** "Line 81: The paper does not include any simulations by using the artificial future data (different emission scenarios) to compare the model adaptability against climate change. So this is not fair to have such an inference. If you want to declare that the models are not robust in extreme event predictions, please cite several climate-related studies (to better link the connection between the climate-change and extreme events)."

**Response.** We agree that it is not appropriate to provide inferences on the models' suitability for climate change predictions. We removed the sentences concerned.

**Modification to manuscript.**

- **Page 3. Line 88.** The sentence was removed: "- Is one approach better suited for climate change predictions?"

- **Page 33. Line 678.** The paragraph was removed: "Given appropriate input data, both models are suited for climate change predictions, as they showed accurate simulations with long calibration periods. However, the operator should keep in mind that the models (i) have difficulty in reproducing extreme events, which are likely to occur more frequently, and (ii) can only reproduce what they know or have learned during the calibration period and cannot take into account the impact of the evolution of recharge processes on the aquifer functioning."

**1.12 Quoting:** "Line 91: I recommend moving map A.1 to this section."

**Modification to manuscript. Figure 1.** We moved Figure A1 to section 2 (main text, now Figure 1).

**1.13 Quoting:** "Line 174-175: In this sub-section, adding a schematic illustration of the ANN model would be very helpful to better understand the modelling architecture."

**Modification to manuscript. Figure 2.** A figure was added to explain the modelling architecture of the ANN model.

**1.14 Quoting:** "Line 180: Please simply explain why you applied the 1-D convolutional layer approach (you may consider citing previous ANN modelling studies)."

**Modification to manuscript. Page 3. Line 79.** A sentence was added in the introduction to explain why we apply 1D Convolutional Neural Network: "We specifically apply 1D Convolutional Neural Networks (CNNs) because in an earlier study (Wunsch et al., 2022) we were able to demonstrate their high ability to perform karst spring discharge modelling. Furthermore, they have some favourable properties compared to popular recurrent neural networks (e.g. the LSTMs), such as batch-wise training procedure which makes them considerably faster and computationally less expensive."

**1.15 Quoting:** "Line 184: I think you can move the names of the python libraries to the appendix. It is not necessary inside the text."

**Modification to manuscript. Page 40. Line 789.** The names of the Python libraries were moved in the *Acknowledgements* section.

**1.16 Quoting:** "Line 185: Here I see you used the library BayesOpt. Was that library used for the Bayesian uncertainty analysis? Or did you apply an uncertainty analysis to the ANN model predictions? If so, please explain which method you applied. Please give some details about the model assumptions, parameter distributions etc. "

**Response.** Actually, we used the library Bayesian Optimization, not BayesOpt. We apologise for this inaccuracy. We did not perform a Bayesian uncertainty analysis with this package, but used it to optimise the ANN hyperparameters. This optimisation strategy unfortunately was not yet explained in the paper. We therefore added clarifying statements and explanations and thank the reviewer for noticing. Please note that the uncertainty is estimated from the model ensemble with 1000 members, which in the case of the CNN are generated by a combination of different model initializations and runs based on monte carlo dropout.

**Modification to manuscript.**

- **Page 8. Line 193.** A sentence was added: "Besides number of filters and number of neurons in the first dense layers, we optimised the training batch size and the length of the input sequence for each simulation step using the *Bayesian Optimization* library (Nogueira, 2014). The number of minimum and maximum optimisation steps was individually selected for each site and can be found in the provided modelling scripts (Cinkus and Wunsch, 2022)."
- **Page 40. Line 784.** The reference of the Python library was corrected: "[...] Bayesian Optimization (Nogueira, 2014) [...]"

**1.17 Quoting:** "Line 189: You may consider pointing out the functionality of the conceptual models in karst water predictions. What is the main advantage of this modelling approach?"

**Response.** Please see the response of comment 1.6. The main advantage of the conceptual models regarding karst water predictions is now detailed in the introduction.

**1.18 Quoting:** "Line 240: You may consider adding a snow reservoir above the Epikarst in Fig 1a. This would be more suitable for the mountainous catchment."

**Response.** Indeed, it would be more appropriate. However, this option is not available in the KarstMod platform, and appears to be one of the limitations of using a platform for reservoir modelling (similar to the consideration of polje and surface water influence in the Unica model). We added some details about this aspect both in the Aubach and Discussion sections.

**Modification to manuscript.**

- **Page 24. Line 418.** The sentence was changed into: **"**These errors can be either due to (i) a miscalibration of the snow routine, retaining too much water as snow in winter and thus releasing too much in warmer periods, or (ii) the snow dynamics which cannot be taken into account within the KarstMod platform, e.g. by adding a snow storage above the epikarst (Chen et al., 2018)."
- **Page 32. Line 647.** The sentence was changed into: "[...] which may benefit the Qachqouch model; (iii) considering polje and surface water influence in the Unica model; or (iv) considering snow dynamics in the structure of the Aubach model."

**1.19 Quoting:** "Line 261: Please mention how to determine the snow routine parameters (Degree-day factor and melting temperature). You mentioned that you did not make an optimization for the snow parameters, so please cite the relevant literature (e.g. He et.al. 2014)."

**Response.** The parameters of the snow routine are now estimated by model calibration. Please see the response to comment 1.5 for more details.

**1.20 Quoting:** "Line 343: The uncertainty bounds are not easy to see especially for the reservoir model in (a). I think when you select all the behavioural simulations (above the threshold), the uncertainty bound will be much more visible. Then the reader can make a visual comparison between them."

**Response.** This has been improved with the rerun of the models. Please see the responses of comments 1.1 and 1.3 for more details.

**1.21 Quoting:** "Line 347: Again the same problem. Please apply a threshold for the reservoir model, and use all the behavioural runs to obtain the uncertainty bounds."

**Response.** This has been improved with the rerun of the models. Please see the responses of comments 1.1 and 1.3 for more details.

**1.22 Quoting:** "Line 355: Please share the model calibration skills in the table."

**Response.** You are right that the information is interesting to provide for reservoir models. However, the score of an ANN model during the calibration period is not relevant as it corresponds to its learning period. Thus, it seems more appropriate to add this detail in appendix to avoid potential confusion for the reader.

**Modification to manuscript.**

- **Table D1.** The table shows the calibration and validation scores of the reservoir models (with a comparison to the ANN models for the validation period).
- **Page 15. Line 372.** The sentence was changed into: "Their performance – assessed with multiple criteria – are shown in Fig. 6, Table 4, and Table D1."

**1.23 Quoting:** "Line 355: You mentioned that the ANN models require long time series to learn the functionality of the karst system. On the other hand, reservoir models could be calibrated for relatively shorter periods. But, there are some results to be discussed in detail as below:

Lez and Qachqouch springs simulation results support the hypothesis, but the Aubach catchment does not. We expect better calibration skills in the reservoir model in a short calibration period. However, the ANN model outperforms the reservoir model in Aubach. How do you explain this?"

**Response.** While Aubach and Qachqouch springs have similar lengths for the calibration period (nearly six years, and about four years, respectively), Lez spring has a longer calibration period of slightly more than nine years, which we consider not exactly a "short" period. For Qachqouch, it is explained Line 511 that "even when data are available, there is a significant amount of time without (relevant) discharge, for which no input-output relation can be learned." In comparison, Aubach is a very reactive system with a lot of "relevant" discharge, which benefit the training of the ANN model. The difference between the springs' regime of Aubach and Qachqouch can be appreciated in Figure 4. We modified the manuscript to emphasise the aspect of relevant discharge in short time series.

**Modification to manuscript.**

- **Page 27. Line 510.** The sentence was corrected into: "The very short data length is particularly detrimental to the ANN model as the learning period is only about 4 years."
- **Page 27. Line 515.** The sentence was changed into: "This highlights the strength of conceptual modelling to take into account recharge processes and reservoir replenishment, even on a short dataset with long dry periods."
- **Page 31. Line 633.** The sentence was changed into: "In contrast, a very short time series or a short time series with long dry periods can be detrimental for the learning of ANN model, which seems to benefit from medium/long periods of relevant discharge (at least 5 years)."
- **Abstract.** The sentence was changed into: "[...] (ii) reservoir models can provide good results even with few years of relevant discharge in the calibration period, [...]"

**1.24    Quoting:** "Line 389: You can make a representative snow routing even if you do not calibrate the snow parameters. Please cite the relevant literature."

> **Response.** The parameters of the snow routine are now estimated by model calibration. Please see the response to comment 1.5 for more details.

**1.25    Quoting:** "Line 527: You may discuss the uncertainty in the temperature data. Temp data strongly affect the timing of the recharge, especially in snow-covered areas (Aubach case)."

> **Modification to manuscript. Page 29. Line 570.** A sentence was added to discuss the uncertainties related to temperature data in snow-covered areas: "In the case of snow-covered areas, it can result in strong uncertainties on the timing of snow accumulation and melting (Zhang et al., 2016), and therefore the recharge of the aquifer."

**1.26    Quoting:** "Line 547: You may add some other model optimization techniques (e.g. cross-validation, see Wilks 2011)."

> **Response.** We agree with the suggestion and modified the manuscript accordingly.

> **Modification to manuscript. Page 30. Line 597.** The sentence was changed into: "[...] or to use different model optimisation techniques, such as cross-validation (Wilks, 2011)."

**1.27    Quoting:** "Line 559: Please discuss the model's structural adequacy here."

> **Modification to manuscript.**
>
> - **Page 30. Line 607.** Several sentences were added: "For high-flow periods, results slightly favour the ANN approach (except for Qachqouch spring), with consistently accurate volumes and shape and timing (Fig. 6). ANN models also tend to achieve higher flows than reservoir models (Fig. 4); due to the noticeable/strong karstification of the studied systems, the high occurrence of high discharge data may benefit the learning of the ANN models. On the other hand, reservoir models are more dependent on the relevance and the quality of the input data preprocessing, thus can be more affected by the uncertainties presented in Sect. 4.2, especially regarding high flows."
> - **Page 30. Line 614.** Several sentences were added: "For low-flow periods, results slightly favour the reservoir approach (except for Aubach spring), with very good estimation of volumes and only a slight overestimation of the hydrological variability (Fig. 6). The conceptual representation of the aquifer with reservoirs and transfer functions may help to simulate the recharge process (especially for inertial systems): a precipitation event will not directly result in a discharge increase at the spring if the reservoir is not fully saturated. On the other hand, ANN models seem to not always account for the time needed for the aquifer to replenish, inducing wave-like behaviours during medium- and low-flow periods (Fig. D1), which can hinder the simulation of low flows."

**Reviewer 2**

**2.1** **Quoting:** "In Data and study sites section (2.1 to 2.5) (p3. From 85-165): it could be better to explain the reason behind using different potential evapotranspiration (PTE) calculation methods for different basins and the impact of different methods on PTE, if there is any."

> **Response.** We agree that it is better to explain why different methods are being used for calculating potential evapotranspiration. We think that there is no negative impact of using different methods; on the contrary, the methods are carefully chosen to provide the most relevant estimations, taking into account the available meteorological data, the climate of the area and local expert knowledge.
>
> **Modification to manuscript. Page 11. Line 283.** The sentence was changed into: "For reservoir models, evapotranspiration processes were considered using time series of potential evapotranspiration, which were calculated for each site using different methods depending on the available meteorological data, the climate of the area and local expert knowledge."

**2.2** **Quoting:** "In Artificial neural networks section (3.1) (p6. From 175 to 185): Selected ANN model type should better be demonstrated with figures as it is done for reservoir models in the following section. Since CNNs are quite complex deep learning models, it can be difficult to be comprehended even for experienced ANN modelers. Demonstrations of CNN and a little more detailed explanation of the regularization methods to avoid traps such as exploding gradient could enrich the context of the article."

> **Response.** A figure has been added to better explain the functioning of ANN models. Further details have also been added to the description of the ANN modelling approach.
>
> **Modification to manuscript.**
>
> - **Figure 2.** A figure was added to explain the modelling architecture of the ANN model.
> - **Page 8. Line 196.** The sentences were added: "To ensure proper learning, the models are regularised with several measures. Hence, early stopping with a patience of 20 Steps is applied to prevent the model from overfitting. Except Qachqouch, where little data is available, the size of the according stopset ranges between one and four annual cycles (see provided scripts for details). This stopset is considered a part of the calibration period mentioned in Sect. 3.4. Further, dropout ensures robust training and serves as another measure against overfitting. We applied the Adam optimizer for a maximum of 150 to 300 training epochs with an initial learning rate of 0.001 and applied gradient clipping to prevent exploding gradients."

**2.3** **Quoting:** "In Model evaluation section section (3.4) Model evaluation criteria should be better justified. Moreover, using more evaluation criteria can be more rigorous which are usually considered in hydrological modelling studies. For instance, relative volume bias, normalized peak error, root mean square error, nash score (the last two is suggested sice they are also used as cost function)."

**Response.** Thanks for the recommendation, we added the MSE to the list of evaluation criteria as it is the performance metric used for calibrating the ANN and reservoir models. Now the models are evaluated using MSE, modified KGE, Diagnostic Efficiency, $\beta$ for the volume bias, $\gamma$ for the discharge variability and the Pearson correlation coefficient $r$ for the discharge shape and timing. We also explained in more details why we evaluate the results using the modified KGE instead of the Nash-Sutcliffe Efficiency.

**Modification to manuscript.**

- **Page 14. Line 338.** The sentence was added: "The Kling-Gupta Efficiency (KGE) has gained in popularity as it aims to address some limitations of the Nash-Sutcliffe Efficiency (Nash and Sutcliffe, 1970), i.e. (i) the discharge variability is underestimated, (ii) the mean of observed values is not a meaningful benchmark for variables with high variability, and (iii) the normalised bias is dependent to the variability (Gupta et al., 2009, Willmott et al., 2012)."
- **Figure 6. Table 4.** Results now also include the MSE for the evaluation of model performance.

**2.4 Quoting:** "In Introduction section (p. 2) from line 48 to 51, authors said "Distributed models require a lot of data for defining physical parameters and thus can 50 be tough to use in a scarce data context. On the other hand, data-driver models permit studying complex and heterogeneous karst systems without requiring extensive meteorological and system-related data.". However, the difference between distributed and data-driven models (if the authors meant statistical models such as ANN) about the data requirement is not necessarily about the amount of data but rather about the diversity. This phrase can be modified to indicate that distributed models need more diverse data while ANNs need only input and output data."

**Response.** We thank the reviewer for the suggestion. The sentence has been modified to emphasise that distributed models need more diverse data (both with high spatial and temporal resolution), in contrast to lumped parameter models.

**Modification to manuscript.**

- **Page 2. Line 49.** The sentence was changed into: "Distributed models require a lot of diverse data with high spatial and temporal resolution for defining physical parameters and thus can be tough to use in a scarce data context (Hartmann et al., 2014)."
- **Page 2. Line 51.** The sentence was changed into: "On the other hand, data-driven models permit studying complex and heterogeneous karst systems without requiring extensive meteorological and system-related data with high spatial resolution."

**2.5 Quoting:** "In Introduction section (1) (p. 2) from line 57 to 62 references can be given in historical order. In Introduction section (1) (p. 3) in line 74 references can be given in historical order. In Reservoir model section (3.2) (p. 7) in line 74 references can be given in historical order. [References can be homogenized as so...]"

**Modification to manuscript.** All the references were ordered in historical order.

**2.6 Quoting:** "In Source of uncertainties section (4.2) (p.23) from line 518 to 540, authors can mention the expected (if not quantified) amount of uncertainty for each source in their case studies for different basins either by expertise or literature support. Since basins are of different characteristics, it can be interesting to have such information for readers."

**Response.** Such information is unfortunately not available for the basins of this study, either by expertise or literature support. However, in the literature, several authors focused their work on quantifying the uncertainties related to input and observed data in the context of karst aquifers. We have added details about this for each source of uncertainty. To the best of our knowledge, quantitative information about model structure uncertainties in karst environments has never been explicitly mentioned.

**Modification to manuscript.**

- **Page 29. Line 567.** The sentence was added: "McMillan et al. (2018) suggested that uncertainties in precipitation data are about 0–10 % at point scale but can go up to 40 % when considering interpolation uncertainties."
- **Page 29. Line 580.** The sentence was added: "The uncertainties related to discharge measurements are highly dependent on the quality of the gauging station and usually range between 10–40 % (McMillan et al., 2018). Although they are expected to be higher in a karst context (Westerberg et al., 2016), some authors reported uncertainties of about 20 % (Jeannin et al., 2021) or 10–15 % (Katz et al., 2009)."

**Additional modifications to the manuscript**

**3.1 Grammar and error check**

**Modification to manuscript.**

- **Page 28. Line 534.** The sentence was changed into: "[...] which cannot be simulated within the KarstMod platform."
- **Page 9. Line 222.** The sentence was changed into: "Structure of models built using the KarstMod platform [...]"
- **Table 3.** The ending date of the validation period of Lez spring was changed to "2020-12-30" instead of "2020-12-03".

**3.2 Minor improvements**

**Modification to manuscript.**

- **Page 2. Line 64.** The sentence was changed into: "[...] only few studies have been conducted on the comparison of both approaches in karst environments (Kong A Siou et al., 2014; Sezen et al., 2019; Jeannin et al., 2021)."
- **Figure 1.** The references in the caption for the carbonate outcrops was updated to "Goldscheider et al. (2020)".
- **Page 29. Line 566.** The reference was updated to "Hohmann et al. (2021)", which corresponds to the published version of the manuscript.
- **Page 28. Line 538.** A reference was added about the flooding of poljes at Unica springs: "[...] that influences the monitoring station (Mayaud et al., 2022)."
- **Page 25. Line 429.** A reference was added about the spatial heterogeneity of snow processes in karst modelling: "As these preprocessings cannot really catch the spatial heterogeneity of complex snow processes, they strongly limit the model performance (Çallı et al., 2022)."
- **Page 33. Line 688.** The sentence was removed as the module is now implemented: "Implementation of a meteorological module in KarstMod for directly calibrating the parameters of the snow routine is besides under implementation."

**References**

Addor, N. and Melsen, L. A.: Legacy, Rather Than Adequacy, Drives the Selection of Hydrological Models, Water Resources Research, 55, 378–390, https://doi.org/10.1029/2018WR022958, 2019.

Goldscheider, N., Chen, Z., Auler, A. S., Bakalowicz, M., Broda, S., Drew, D., Hartmann, J., Jiang, G., Moosdorf, N., Stevanovic, Z., and Veni, G.: Global distribution of carbonate rocks and karst water resources, Hydrogeol J, 28, 1661–1677, https://doi.org/10.1007/s10040-020-02139-5, 2020.

Hohmann, C., Kirchengast, G., O, S., Rieger, W., and Foelsche, U.: Small Catchment Runoff Sensitivity to Station Density and Spatial Interpolation: Hydrological Modeling of Heavy Rainfall Using a Dense Rain Gauge Network, Water, 13, 1381, https://doi.org/10.3390/w13101381, 2021.

Jeannin, P.-Y., Artigue, G., Butscher, C., Chang, Y., Charlier, J.-B., Duran, L., Gill, L., Hartmann, A., Johannet, A., Jourde, H., Kavousi, A., Liesch, T., Liu, Y., Lüthi, M., Malard, A., Mazzilli, N., Pardo-Igúzquiza, E., Thiéry, D., Reimann, T., Schuler, P., Wöhling, T., and Wunsch, A.: Karst modelling challenge 1: Results of hydrological modelling, J. Hydrol., 600, 126508, https://doi.org/10.1016/j.jhydrol.2021.126508, 2021.

Katz, B. G., Sepulveda, A. A., and Verdi, R. J.: Estimating Nitrogen Loading to Ground Water and Assessing Vulnerability to Nitrate Contamination in a Large Karstic Springs Basin, Florida1, JAWRA Journal of the American Water Resources Association, 45, 607–627, https://doi.org/10.1111/j.1752-1688.2009.00309.x, 2009.

Mayaud, C., Kogovšek, B., Gabrovšek, F., Blatnik, M., Petrič, M., and Ravbar, N.: Deciphering the water balance of poljes: example of Planinsko Polje (Slovenia), Acta Carsologica, 51, https://doi.org/10.3986/ac.v51i2.11029, 2022.

McMillan, H. K., Westerberg, I. K., and Krueger, T.: Hydrological data uncertainty and its implications, WIREs Water, 5, e1319, https://doi.org/10.1002/wat2.1319, 2018.

Nash, J. E. and Sutcliffe, J.: River flow forecasting through conceptual models: Part 1. A discussion of principles., J. Hydrol., 10, 282–290, 1970.

Westerberg, I. K., Wagener, T., Coxon, G., McMillan, H. K., Castellarin, A., Montanari, A., and Freer, J.: Uncertainty in hydrological signatures for gauged and ungauged catchments, Water Resources Research, 52, 1847–1865, https://doi.org/10.1002/2015WR017635, 2016.

Wilks, D. S.: Statistical Forecasting, in: International Geophysics, vol. 100, Elsevier, 215–300, https://doi.org/10.1016/B978-0-12-385022-5.00007-5, 2011.

Willmott, C. J., Robeson, S. M., and Matsuura, K.: A refined index of model performance, Intern. J. Climatol., 32, 2088–2094, https://doi.org/10.1002/joc.2419, 2012.

Zhang, J. L., Li, Y. P., Huang, G. H., Wang, C. X., and Cheng, G. H.: Evaluation of Uncertainties in Input Data and Parameters of a Hydrological Model Using a Bayesian Framework: A Case Study of a Snowmelt–Precipitation-Driven Watershed, Journal of Hydrometeorology, 17, 2333–2350, https://doi.org/10.1175/JHM-D-15-0236.1, 2016.